# Smoking as a risk factor for lung cancer in women and men: a systematic review and meta-analysis

Linda M O'Keeffe,[1,2] Gemma Taylor,[1,2,3,4] Rachel R Huxley,[5,6] Paul Mitchell,[7] Mark Woodward,[6,8,9] Sanne A E Peters[8]

LMO'K and GT contributed equally.

For numbered affiliations see end of article.

**Correspondence to**
Dr Linda M O'Keeffe;
linda.okeeffe@bristol.ac.uk

## ABSTRACT

**Objectives** To investigate the sex-specific association between smoking and lung cancer.

**Design** Systematic review and meta-analysis.

**Data sources** We searched PubMed and EMBASE from 1 January 1999 to 15 April 2016 for cohort studies. Cohort studies before 1 January 1999 were retrieved from a previous meta-analysis. Individual participant data from three sources were also available to supplement analyses of published literature.

**Eligibility criteria for selecting studies** Cohort studies reporting the sex-specific relative risk (RR) of lung cancer associated with smoking.

**Results** Data from 29 studies representing 99 cohort studies, 7 million individuals and >50 000 incident lung cancer cases were included. The sex-specific RRs and their ratio comparing women with men were pooled using random-effects meta-analysis with inverse-variance weighting. The pooled multiple-adjusted lung cancer RR was 6.99 (95% Confidence Interval (CI) 5.09 to 9.59) in women and 7.33 (95% CI 4.90 to 10.96) in men. The pooled ratio of the RRs was 0.92 (95% CI 0.72 to 1.16; $I^2$=89%; p<0.001), with no evidence of publication bias or differences across major pre-defined participant and study subtypes. The women-to-men ratio of RRs was 0.99 (95% CI 0.65 to 1.52), 1.11 (95% CI 0.75 to 1.64) and 0.94 (95% CI 0.69 to 1.30), for light, moderate and heavy smoking, respectively.

**Conclusions** Smoking yields similar risks of lung cancer in women compared with men. However, these data may underestimate the true risks of lung cancer among women, as the smoking epidemic has not yet reached full maturity in women. Continued efforts to measure the sex-specific association of smoking and lung cancer are required.

## INTRODUCTION

Lung cancer is the leading cause of cancer death worldwide with 1.7 million global deaths attributed to cigarette smoking in 2015.[1] Tobacco use is the leading cause of lung cancer; 55% of lung cancer deaths in women and over 70% of lung cancer deaths in men are due to smoking.[1] These global estimates, however, mask major differences in smoking prevalence in men and women

## Strengths and limitations of this study

► Evidence on the sex-specific association of smoking and lung cancer was meta-analysed in over 7 million participants across 99 cohort studies.

► Several subgroup analyses were performed to examine the robustness of findings across different population subgroups.

► However, the smoking epidemic is not yet fully mature in women and risks of lung cancer in women may still be underestimated.

► Detailed data on smoking behaviour and data on specific subtypes of lung cancer were not available.

across populations, with rates below 5% for women in most Asian and African countries to 40% and above for men in many parts of Asia and Eastern Europe.[2] In addition, smoking behaviour varies significantly by sex. For example, compared with women, men smoke more cigars and pipes,[3] take puffs of longer duration and leave shorter butts,[4] which each could potentially predispose them to greater risks of smoking-related lung cancer. Substantial physiological differences between the sexes may also result in sex differences in the effects of smoking, particularly for women. For example, compared with men, women have a smaller lung size and different airway behaviour,[5] which may increase their susceptibility to lung cancer at lower levels of smoking. A recent meta-analysis showed that cigarette smoking confers a greater coronary hazard in women compared with men, which suggests the possibility that this may also be true for the risk of smoking-related lung cancer.[6]

A study of 50-year trends in smoking-related mortality in the USA found that the relative risks of smoking-related lung cancer mortality were higher in men than women.[7] However, this sex difference was only apparent in the oldest cohorts with the longest follow-up, possibly reflecting greater cumulative tobacco

exposure in men than in women. In contrast, a recent study in Korea, a population where smoking patterns continue to differ between the sexes, suggested that sex differences in the impact of smoking on lung cancer risk exist and differ by histological subtype.[8] Analyses of a large UK primary care database showed that moderate and heavy smoking more strongly increase the risks of lung cancer in women than in men.[9]

Two recent meta-analyses examined the sex-specific association between smoking and lung cancer. In the most recent of these, men were found to have a greater risk of lung cancer associated with smoking compared with women.[10] However, virtually all data were from historical case–control studies, which have several limitations, and the three included prospective studies provided contradictory results. While a previous meta-analysis by Lee *et al*[11] included 287 cohort and case–control studies and provided sex-specific estimates, single-sex cohorts were also included, sex differences in the smoking-related risk of lung cancer were not formally compared within studies, and only studies published up to 1999 were included.

To resolve this uncertainty, we performed a systematic review and meta-analysis of prospective cohort studies published to date on the sex-specific association of smoking with the risk of fatal and non-fatal lung cancer. Our systematic review builds on these previous meta-analyses by adding literature from 1999 onwards and restricting the analyses to cohort studies, which are less prone to bias than case–control studies. In addition, we perform several predefined subgroup analyses which have not been performed in meta-analyses of cohort studies included in previous reviews and supplement our findings with results from three sources of individual participant data (IPD), not published previously. An important a priori consideration is the substantial sex difference in the maturity of the smoking epidemic with men being at a more advanced stage than women in most parts of the world.[2] This would be expected to translate into lower relative risk (RR) estimates for lung cancer in women than in men. Hence, the null hypothesis that smoking confers the same lung cancer hazard in both women and men, would be met if the ratio of the RRs (RRRs) for lung cancer (women:men) was less than unity (reflecting a greater hazard in men than women). However, if the RRRs were found to be unity (or higher) then this would suggest a greater hazard associated with tobacco exposure in women than in men.

## METHODS
### Search strategy
This review was conducted using a predefined protocol and in accordance to the Meta-analysis Of Observational Studies in Epidemiology guidelines (online supplementary eappendix 1). We systematically searched PubMed and EMBASE for studies published between 1 January 1999 and 15 April 2016 that reported on the relationship between smoking and lung cancer in men and women

from a general population. The computer-based searches combined medical subject headings and free-text terms related to 'tobacco/smoking', 'cancer', 'sex' and 'cohort studies'. The full search criteria are available in online supplementary eappendix 2. Articles published before 1 January 1999 were retrieved from a previous systematic review.[11] The reference lists of all relevant original research and review articles were scanned to capture missed studies. Two authors (LMOK and GT) independently conducted the screening of studies and any disagreement was mediated by a third author (SAEP).

### Data extraction
Data were extracted, in duplicate, from studies deemed to meet the eligibility criteria. These included details on general study characteristics (study name, duration of follow-up, year of publication), information about the studied population (prevalence of smoking, mean age, number of men and women, incidence of lung cancer, whether lung cancer was fatal or non-fatal and level of adjustment for covariates). We extracted sex-specific adjusted measures of RR and 95% confidence intervals (CIs).

### Study selection
Observational cohort studies were included if they reported sex-specific RRs or equivalent, on the relationship between smoking and lung cancer. Studies were excluded if the variability around the point estimate was not reported, if they had not been adjusted for at least age, or if the study was performed in a population selected on the basis of prior lung cancer or another major underlying chronic disease. In the case of duplicate reports from the same study, the report with the longest follow-up or the highest number of cases was included. IPD from studies available to the authors were also used; the Asia Pacific Cohort Studies Collaboration (APCSC), the National Health and Nutrition Examination Survey III (NHANES III) and the Scottish Heart Health Extended Cohort Study (SHHEC). The Newcastle-Ottawa Scale assessment (NOS) was used to assess the methodological quality of all included studies, on a 9-point scale (online supplementary eappendix 3 and etable 1).[12]

### Meta-analysis
The primary analysis was a comparison of the sex-specific RR of lung cancer (fatal or non-fatal) in current smokers versus non-smokers (defined either as former or never smokers). For each study, we obtained the natural log of the sex-specific RRs and calculated the differences. The differences were pooled across studies using random-effects meta-analysis which allows the RR of lung cancer to vary from study to study, weighted by the inverse of the variances of the log RRs and then back-transformed to obtain the pooled women-to-men RRRs. The SE of the log RRR was calculated as the square root of the sum of the variance of the two sex-specific log RRs for each study. Pooled RRRs were computed separately for

studies with only age-adjusted estimates and then for those studies with multiple-adjusted estimates. The set of multiple adjustments made was allowed to vary by study, but had to include at least one other risk factor in addition to age. The I² statistic was used to estimate the percentage of variability across studies due to between-study heterogeneity. The presence of publication bias was graphically examined using contour funnel plots, plotting the natural log of the RRR against its SE and tested using Begg's test. Predefined subgroup analyses were conducted to obtain the adjusted RRRs by study region (Asia or non-Asia and Asia, Europe, USA, and Australia and New Zealand (ANZ)), year of study baseline (pre-1985 or post-1985), study endpoint (fatal only or fatal and non-fatal combined), number of cigarettes smoked per day (>0 to 10, 10–20, >20), study quality ≤6 vs >6 points) and follow-up time (≤10 vs >10 years). Random-effects meta-analyses were used for all subgroup analyses and differences between subgroups were examined using meta-regression. To include the largest number of studies

available, we combined the age-adjusted and multiple-adjusted estimates, taking the maximum adjustment set available. In secondary analyses, we obtained the sex-specific RRs and RRRs comparing former smokers to never smokers and performed the same set of subgroup analyses. All analyses were performed using Stata V.12.0.

## Patient and public involvement

There were no patients or applicable public involved in this review.

## RESULTS

Of the 9519 unique records that were identified through the systematic search, 227 qualified for full-text evaluation (figure 1). Of these, 25 separate studies provided information about sex differences in the association between smoking and lung cancer. This database was extended with IPD from APCSC (separately for Asia and ANZ), NHANES III and SHHEC leading to a total of 29

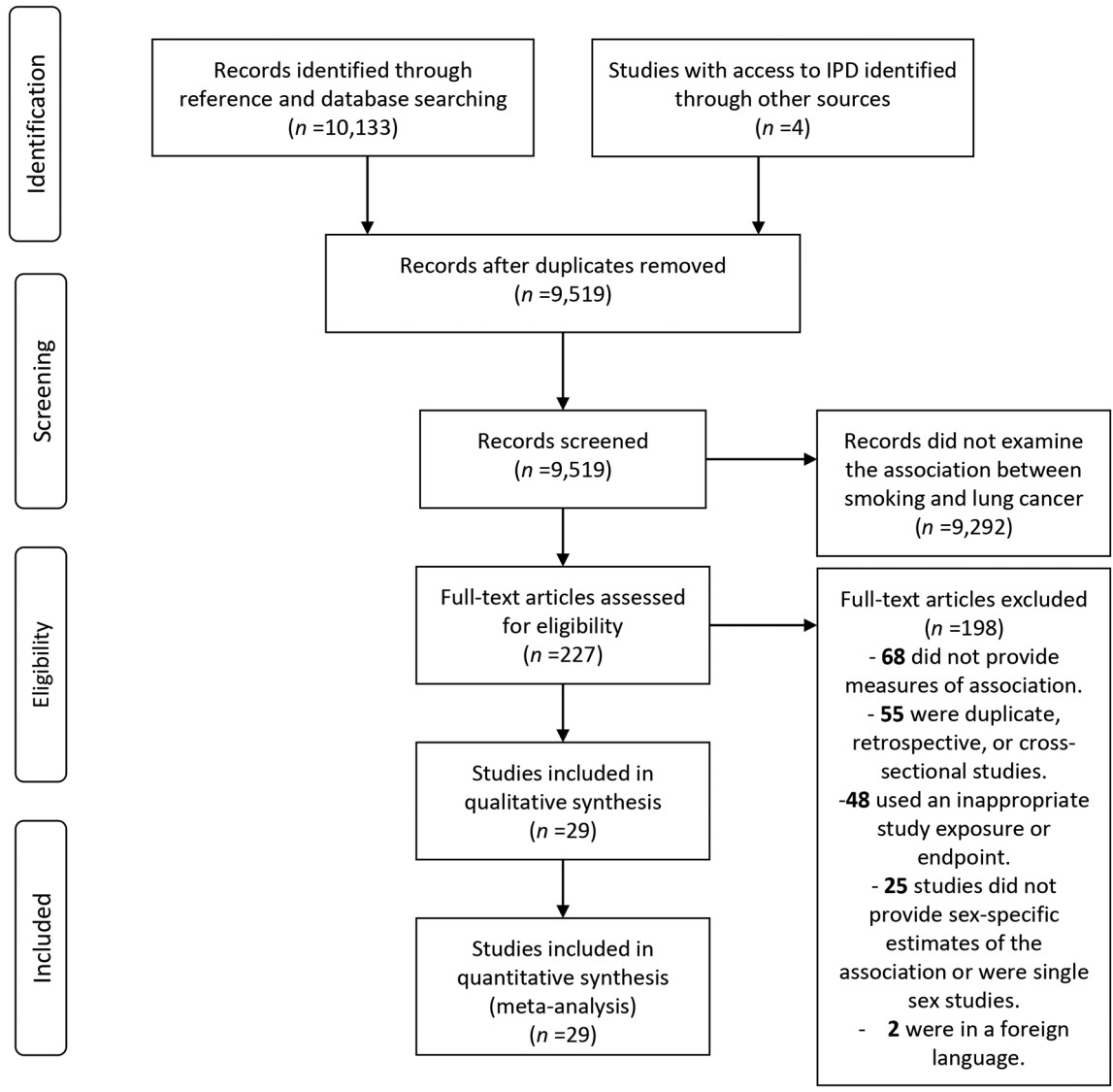

**Figure 1** Flow chart of study selection. IPD, individual participant data.

individual estimates, representing a total of 99 cohort studies available for meta-analysis.

The characteristics of the included studies are described in table 1. Overall, data were available from 99 cohorts, including 7 113 303 individuals (46% women)—not accounting for two cohorts that used Census data—and at least 51 161 incident cases of lung cancer (31% women). Forty-six cohorts were from Asia (61% of the individuals), 6 were from the USA (28%), 37 were from Europe (10%) and 10 were from ANZ (1%). Of 29 studies, 4 studies had a quality score of 5 out of 9, 9 studies had a score of 6, 12 studies had a score of 7 and 4 studies with a score of 8 (online supplementary etable 1).

Eighteen studies reported on the prevalence of smoking, which varied widely by study, region and sex. The prevalence of smoking ranged from 1% to 47% in women and from 1% to 70% in men. In all but two studies, the prevalence of smoking was higher in men than women, especially in Asia where typically less than 10% of women were smokers compared with over 50% of men. Smoking cessation rates were also higher among men (7%–61%) than women (<1%–39%).

### Risk of lung cancer in current smokers versus non-smokers

Compared with non-smoking, current smoking was associated with an age-adjusted RR of lung cancer of 7.48 (95% CI 5.29 to 10.60) in women and 8.78 (95% CI 6.13 to 12.57) in men (table 2 and online supplementary efigure 1). The pooled age-adjusted women-to-men RRR was 0.81 (95% CI 0.62 to 1.04), with substantial between-study heterogeneity ($I^2$=86%; p<0.001) (table 2 and online supplementary efigure 2). The multiple-adjusted RR of lung cancer associated with current smoking was 6.99 (95% CI 5.09 to 9.59) in women and 7.33 (95% CI 4.90 to 10.96) in men (table 2 and figure 2). The corresponding RRR was 0.92 (95% CI 0.72 to 1.16) and between-study heterogeneity was substantial ($I^2$=89%; p<0.001) (table 2 and figure 3). There was no evidence of publication bias based on the Begg's test (p=0.75) (online supplementary efigure 3).

The sex difference in the risk of smoking-related lung cancer in our main analysis did not differ in subgroup analyses stratified by the women-to-men ratio of current smokers (p=0.90), women-to-men ratio of lung cancer incidence in the studies (p=0.64), year of study baseline (p=0.66), study endpoint (p=0.21) or study region (p=0.73) (table 3). The sex difference in the risk of smoking-related lung cancer in our main analysis also did not differ by follow-up time (p=0.83) or study quality (p=0.69). The RRR was 0.93 (95% CI 0.72 to 1.20) for studies from Asia and 0.87 (95% CI 0.66 to 1.14) for studies from USA, Europe or ANZ.

The risk of smoking-related lung cancer increased according to the number of cigarettes smoked per day in both sexes (table 2). In women, the RRs were 5.30 (95% CI 3.52 to 7.97), 10.67 (95% CI 7.43 to 15.33) and 17.09 (95% CI 12.11 to 24.11) across subgroups of <10, 10 to 20 and >20 cigarettes per day versus non-smoking,

respectively. Corresponding RRs in men were 4.97 (95% CI 2.74 to 9.03), 8.93 (95% CI 4.90 to 16.28) and 14.61 (95% CI 8.33 to 25.59), respectively. The RRRs in these subgroups were 0.99 (95% CI 0.65 to 1.52), 1.11 (95% CI 0.75 to 1.64) and 0.94 (95% CI 0.69 to 1.30), respectively.

### Risk of lung cancer in former smokers versus never smokers

Data from 89 cohorts, including 6 006 725 individuals and 38 244 cases of lung cancer, reported on the risk of lung cancer in former smokers compared with never smokers. The age-adjusted RR of lung cancer associated with former smoking was 2.82 (95% CI 2.25 to 3.54) in women and 3.01 (95% CI 2.23 to 4.08) in men (table 2 and online supplementary efigure 4); the age-adjusted RRR was 0.88 (95% CI 0.69 to 1.14) ($I^2$=64%; p<0.001) (table 2 and online supplementary efigure 5). The corresponding multiple-adjusted RRs were 3.14 (95% CI 2.45 to 4.03) in women and 3.13 (95% CI 2.06 to 4.76) in men (table 2 and online supplementary efigure 6). There was no statistical evidence that the effects of smoking cessation on risk of lung cancer differed between the sexes; the multiple-adjusted RRR was 0.89 (95% CI 0.69 to 1.13) ($I^2$=69%; p<0.001) (table 2 and online supplementary efigure 7). There was no evidence that the RRR differed across various subgroup analyses (table 3).

### DISCUSSION

In this systematic review and meta-analysis, comprising data from more than 7 million participants, 99 cohort studies and over 50 000 incident cases of lung cancer, there was no evidence for a difference in the risk of smoking-related lung cancer in women compared with men. This was true across a range of subgroup and sensitivity analyses. However, as smoking prevalence and intensity were higher in men compared with women in most studies included in this analysis, there may yet be an unrealised sex difference in the risk of smoking-related lung cancer that will only become fully manifest as the smoking epidemic reaches full maturity in women.[2]

The sevenfold higher RRs of lung cancer associated with smoking found in the present meta-analysis are considerably smaller than the 20-fold increased risks reported in the Million Women's Study [13] and the British Doctors Study.[14] Both of these studies had the advantage of capturing smoking-related risks in populations that had smoked for long enough for the effects to become fully manifest, highlighting the importance of taking into consideration the stage of the tobacco epidemic in each sex. The lack of any appreciable sex difference in the RRs of lung cancer is surprising given men's greater cumulative exposure to smoking, in most populations, compared with women. In addition, men have a greater exposure to other risk factors for lung cancer including occupational carcinogens.[15] Men also smoke more cigars and pipes,[3] take longer puffs of longer duration and leave shorter butts compared with women.[4] Hence, it may be

**Table 1** Characteristics of included studies

| Study name* | Location | Age range, (year) | Baseline year | Follow-up, (years) | NOS score (points) | N (%) women | N (%) lung cancer in women | Fatal/non-fatal | Current smoker, % W/M | | Former smoker, % W/M | | Maximum available adjustment |
|---|---|---|---|---|---|---|---|---|---|---|---|---|---|
| APCSC—ANZ[31] (8 cohorts) | ANZ | 20–107 | 1961–1993 | 10 | 6 | 87130 (52) | 501 (28) | F | 7 | 11 | 11 | 18 | Age, education, BMI |
| APCSC—Asia[31] (33 cohorts) | Asia | 20–104 | 1989–1996 | 7 | 6 | 476755 (35) | 1275 (19) | F | 2 | 59 | <1 | 7 | Age, education, BMI |
| ARIC[32] (1 cohort) | USA | 45–64 | 1987–1989 | 20 | 8 | 14610 (54) | 470 (37) | F/NF | 25 | 44 | 22 | 28 | Age, race |
| China Kadoorie Biobank[33 34] (1 cohort) | China | 35–74 | 2004–2008 | 7 | 7 | 512891 (59) | 1953 (37) | F | 3 | 61 | 1 | 15 | Age, area, education, alcohol |
| China National Hypertension Survey[35] (1 cohort) | China | 15+ | 1991 | 8 | 7 | 155131 (51) | NR | F | 14 | 63 | NR | NR | Age, education, region, HT, overweight/obesity, alcohol, PA, DM |
| Copenhagen Cohort Studies[36] (3 cohorts) | Denmark | 20+ | 1964–1992 | 14 | 6 | 30874 (44) | 867 (23) | F/NF | NR | NR | NR | NR | Age |
| CPS I[37] (1 cohort) | USA | 55+ | 1959 | 6 | 5 | 518982 (65) | 1293 (21) | F | 15 | 40 | 4 | 17 | Age, education, race |
| CPS II[37] (1 cohort) | USA | 55+ | 1982 | 6 | 5 | 746485 (61) | 4957 (36) | F | 18 | 24 | 21 | 43 | Age, education, race |
| EHS[38 39] (1 cohort) | China | 65+ | 1998–2000 | 11 | 7 | 65510 (66) | 1096 (27) | F | 4 | 10 | 8 | 20 | Age, education, alcohol, depression, health status, social security assistance, housing type, expenditure |
| EPIC[40] (23 cohorts) | Europe | 30–70 | 1992–2000 | 11 | 8 | 441211 (70) | 2995 (49) | F/NF | 21 | 27 | 25 | 38 | Age, education, BMI, alcohol, PA, energy intake, diet |
| JPHC,[41] JACC,[37 42 43] TPCS[37 42 43] (3 cohorts) | Japan | 40–79 | 1983–1994 | 10 | 6 | 296836 (53) | NR | F | 8 | 54 | 2 | 25 | Age |
| Korean Cancer Prevention Study[44] (1 cohort) | Korea | 30–95 | 1993–1995 | 9 | 6 | 1212906 (32) | 4238 (14) | F | 5 | 57 | 3 | 23 | Age |
| Korean National Health Insurance Service[8] (1 cohort) | Korea | 20+ | 1998–1999 | 10 | 7 | 1355891 (31) | 6491 (15) | F/NF | 1 | 56 | 1 | 13 | Age, BMI, alcohol, PA |
| Malmo Preventive Project[45] (1 cohort) | Sweden | 27–61 | 1974–1992 | 24 | 5 | 33346 (33) | 436 (21) | F | 36 | 49 | 19 | 27 | Age, FEV, SES, marital status |
| Migrant Study[46] (1 cohort) | Norway | 33–72 | 1964–1965 | 21 | 7 | 26126 (55) | 435 (23) | F | 26 | 46 | 8 | 28 | Age |
| New Zealand Census 1981[47] (1 cohort) | New Zealand | 25+ | 1981 | 5 | 7 | NR | 4188 (28) | F/NF | NR | NR | NR | NR | Age, race |
| New Zealand Census 1996[47] (1 cohort) | New Zealand | 25+ | 1996 | 5 | 7 | NR | 4467 (44) | F/NF | NR | NR | NR | NR | Age, race |
| NHANES III[47] (1 cohort) | USA | 18–90 | 1988–1994 | 13 | 7 | 20006 (53) | 320 (36) | F | 21 | 30 | 17 | 32 | Age, education, BMI |
| NHIS[48] (1 cohort) | USA | 25+ | 1997–2004 | 7 | 7 | 202248 (56) | 1223 (44) | F | 21 | 26 | 20 | 29 | Age, education, BMI, alcohol |

Continued

**Table 1** Continued

| Study name* | Location | Age range, (year) | Baseline year | Follow-up, (years) | NOS score (points) | N (%) women | N (%) lung cancer in women | Fatal/non-fatal | Current smoker, % W/M | | Former smoker, % W/M | | Maximum available adjustment |
|---|---|---|---|---|---|---|---|---|---|---|---|---|---|
| | | | | | | | | | W | M | W | M | |
| NIH-AARP[7 49] (1 cohort) | USA | 50–71 | 1995–1996 | 11 | 7 | 452 131 (41) | 9381 (37) | F/NF | 17 | 13 | 39 | 61 | Age, education, alcohol, ethnicity |
| Norwegian Counties Study[50–52] (1 cohort) | Norway | 20–49 | 1974–1978 | 23 | 6 | 48 682 (48) | 686 (35) | F | 43 | 50 | 11 | 18 | Age, SBP, TC, TG, PA, BMI, height, sickness leave, disability pension, family history of CHD |
| Renfrew/Paisley Study[53] (1 cohort) | Scotland | 45–64 | 1972–1976 | 20 | 6 | 15 393 (54) | NR | F | 47 | 59 | 7 | 25 | Age |
| Reykjavik Study[54] (1 cohort) | Iceland | 32–60 | 1967 | 27 | 7 | 22 946 (50) | 472 (42) | F/NF | 39 | 30 | 15 | 24 | Age |
| Shanghai Health Study[55] (2 cohorts) | China | 40–74 | 1997–2006 | 10 | 7 | 134 335 (54) | 839 (49) | F | 3 | 70 | NR | NR | Age, education, BMI, alcohol, PA, income |
| SHHEC[1 cohort] | Scotland | 25–75 | 1984–1987 | 23 | 6 | 17 731 (51) | 558 (38) | F | 40 | 51 | 21 | 27 | Age, education, BMI |
| Singapore Chinese Health Study[56] (1 cohort) | Singapore | 45–74 | 1993–1998 | 12 | 8 | 61 320 (55) | 905 (31) | F/NF | 6 | 36 | 3 | 21 | Age, dialect, year of recruitment, education, alcohol, PA |
| Swedish Smoking Habit Survey[57 58] (1 cohort) | Sweden | 18–69 | 1963 | 33 | 8 | 41 544 (60) | 442 (45) | F | 18 | 27 | 5 | 23 | Age, area |
| Swiss National Cohort[59] (4 cohorts) | Switzerland | 14–99 | 1977–1993 | 19 | 5 | 35 703 (53) | 426 (29) | F | 27 | 43 | 13 | 24 | Age, survey, education, alcohol, PA, civil status, nationality, diet |
| Wen et al[60] (2 cohorts) | Taiwan | 35+ | 1982–1992 | 20 | 5 | 86 580 (39) | 247 (2) | F | 1 | 41 | <1 | 11 | Age |

*Note that where several studies are cited for a single cohort, data may be extracted from multiple studies if all data required is not available in the most up-to-date and relevant study.
ANZ, Australia and New Zealand; ARIC, Atherosclerosis Risk in Communities study; APCSC, Asia Pacific Cohort Studies Collaboration; BMI, body mass index; CHD, coronary heart disease; CPS, Cancer Prevention Study; DM, diabetes mellitus; EHS, Elderly Health Services; EPIC, European Prospective Investigation into Cancer; F, fatal; FEV, forced expiratory volume; JACC, Japan Collaborative Cohort Study; JPHC, Japan Public Health CentreStudy; NF, non-fatal; NHANES III, National Health And Nutrition Examination Survey III; NHIS, National Health Interview Survey; NIH-AARP, National Institutes of Health American Association of Retired Persons Diet and Health Study; NOS, Newcastle-Ottawa Scale; NR, not reported; PA, physical activity; SBP, systolic blood pressure; SES, socioeconomic status; SHHEC, Scottish Heart Health Extended Cohort Study; TC, total cholesterol; TG, triglycerides; TPCS, Three-Prefecture Cohort Study.

**Table 2** Sex-specific pooled relative risks (RR) and ratio of relative risks (RRR) for lung cancer associated with smoking

| | RR in women | RR in men | RRR |
|---|---|---|---|
| **Age adjusted** | | | |
| Former versus never | 2.82 (2.25 to 3.54) | 3.01 (2.23 to 4.08) | 0.88 (0.69 to 1.14) |
| Current versus not | 7.48 (5.29 to 10.60) | 8.78 (6.13 to 12.57) | 0.81 (0.62 to 1.04) |
| **Multiple adjusted** | | | |
| Former versus never | 3.14 (2.45 to 4.03) | 3.13 (2.06 to 4.76) | 0.89 (0.69 to 1.13) |
| Current versus not | 6.99 (5.09 to 9.59) | 7.33 (4.90 to 10.96) | 0.92 (0.72 to 1.16) |
| **Maximum adjusted** | | | |
| Former versus never | 2.92 (2.35 to 3.63) | 3.08 (2.31 to 4.11) | 0.86 (0.71 to 1.05) |
| Current versus not | 7.32 (5.58 to 9.61) | 8.05 (5.90 to 10.98) | 0.89 (0.73 to 1.08) |
| **Cigarettes per day among current smokers versus never (maximum available adjusted)** | | | |
| 10 or less | 5.30 (3.52 to 7.97) | 4.97 (2.74 to 9.03) | 0.99 (0.65 to 1.52) |
| 10 to 19 | 10.67 (7.43 to 15.33) | 8.93 (4.90 to 16.28) | 1.11 (0.75 to 1.64) |
| 20 or more | 17.09 (12.11 to 24.11) | 14.61 (8.33 to 25.59) | 0.94 (0.69 to 1.30) |

Multiple adjusted includes anything that adjusted for more than just age. Maximum available adjustment refers to the most adjustments provided in the study. For some studies, this would have been age adjusted whereas other studies adjusted for more factors than age only (ie, multiple adjusted). These covariates are listed in table 1.

reasonable to surmise that the RR estimates of smoking-related lung cancer in women may eventually exceed those of men, once cumulative exposure to smoking in women is comparable to that in men. In a previous meta-analysis, using similar methodology, we found that smoking conferred a 25% greater RR of coronary heart disease (CHD) in women than in men. Two possible explanations for why a similar pattern is not observed for lung cancer are that, first, the lag-time between smoking and CHD is considerably shorter than for lung cancer,[16] and second the pathways by which smoking increases risk are different between CHD and lung cancer.

Although not assessed in this analysis, evidence suggests that there are sex differences in the pattern of lung cancer among never smokers, with a higher prevalence of lung cancer among never-smoking women than never-smoking men.[17 18] A US study among 500000 people found a 30% higher incidence of lung cancer in women never smokers compared with men never smokers.[19] An Australian study found the proportion of patients with lung cancer who had never smoked was approximately 18% in women and 3% of men.[20] The reasons for this sex difference are not clear, but women may have increased exposure to environmental tobacco smoke[21] or other environmental carcinogens such as indoor air pollution[22] or sex-related differences in the metabolism of environmental carcinogens. The possibility of greater exposure to environmental tobacco smoke and other environmental carcinogens in women compared with men could have resulted in a greater underestimation of the association between smoking and lung cancer in women than men. This, in turn, could have impacted the sex difference in risk of smoking-related lung cancer reported in this study.

Our study has several strengths including restriction to cohort studies which provide more robust evidence of the associations compared with case–control studies. Differences between case–control and cohort studies may also explain why a previous meta-analysis of case–control studies (which included only three cohort studies) showed a higher RR of lung cancer in men compared with women.[10] Other strengths to our study include an update of findings to include studies published after 1999,[11] with supplementation of published literature with IPD from three established population databases. We have also performed a range of prespecified sensitivity analyses and several subgroup analyses which were not performed in previous meta-analyses. Our results were consistent across regions and irrespective of the women-to-men smoking ratio, suggesting that underestimation of the association of smoking and lung cancer in women due to sex differences in smoking prevalence and under-reporting of smoking is unlikely. This is especially relevant for parts of Asia where the prevalence of smoking in women is typically <10% and where smoking among women remains relatively socially unacceptable. As the up-take of smoking continues among women in countries where significant sex differences in smoking prevalence exist, the sex-specific risks of lung cancer due to smoking may become further apparent. This is also true for Western countries where differences in prevalence between women and men have reduced substantially over time, with prevalence of smoking in younger cohorts of women and men approaching unity.[23] The limitations of this study include heterogeneity across studies in study design, study population, verification of smoking status and outcome ascertainment. Assessment of smoking status differed across studies and was generally self-reported, which may have introduced measurement error.[24] Notably, compared with men, women are more likely to under-report smoking status,

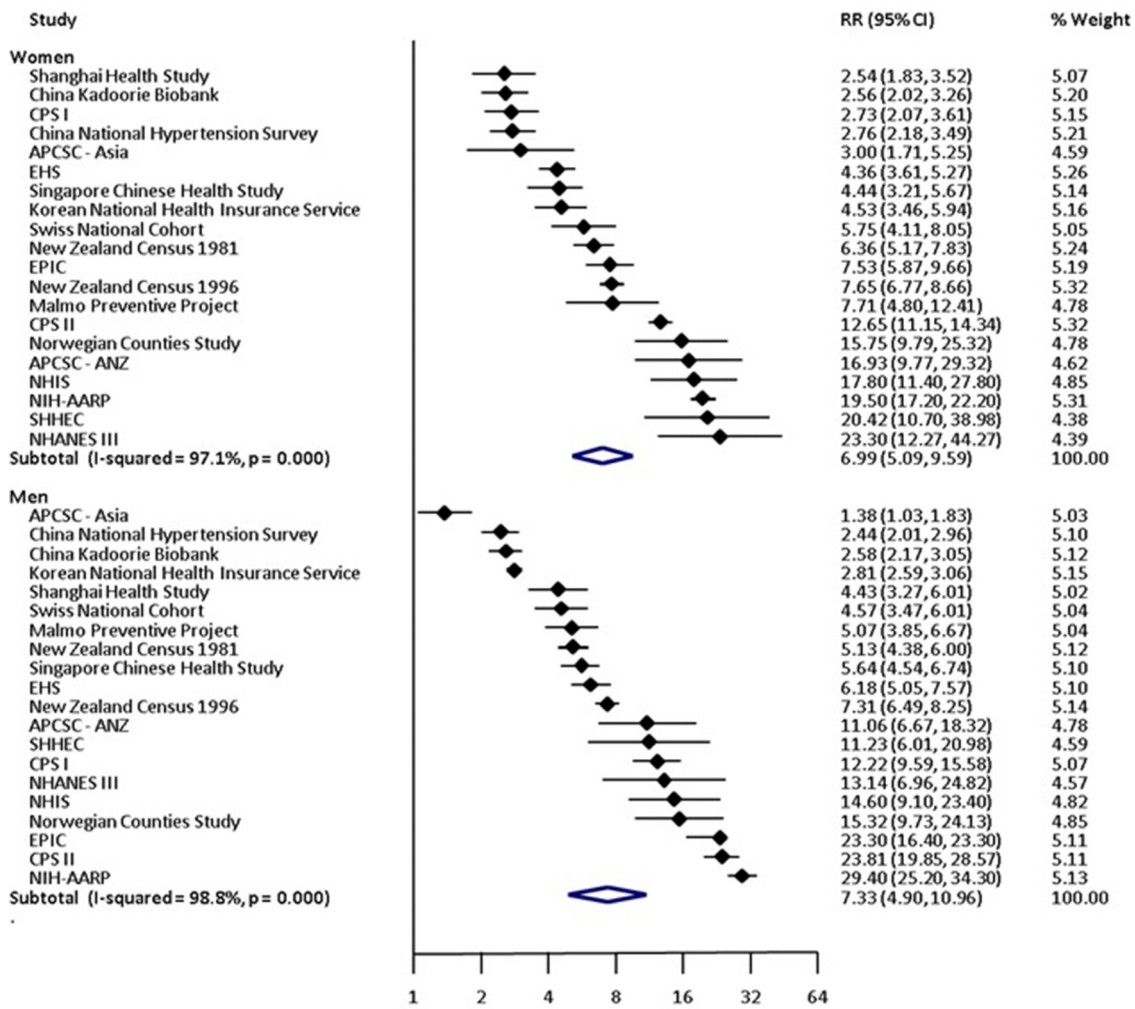

**Figure 2** Multiple-adjusted relative risk (RR) for incident lung cancer in women and men, comparing current smokers to non-smokers. Multiple-adjusted includes anything that adjusted for more than just age. These covariates are listed in table 1. Figures may contain less than 29 studies because we report age-adjusted and multiple-adjusted results separately. Some studies only contributed age-adjusted results whereas others only provided multiple-adjusted results. However, the count of unique studies that contributed to at least one of these analyses is 29. APCSC, Asia Pacific Cohort Studies Collaboration; ARIC, Atherosclerosis Risk in Communities study; CPS, Cancer Prevention Study; EHS, Elderly Health Services; EPIC, European Prospective Investigation into Cancer; JACC, Japan Collaborative Cohort Study; JPHC, Japan Public Health Centre Study; NHANES III, National Health And Nutrition Examination Survey III; NHIS, National Health Interview Survey; NIH-AARP, National Institutes of Health American Association of Retired Persons Diet and Health Study; SHHEC, Scottish Heart Health Extended Cohort Study; TPCS, Three-Prefecture Cohort Study.

and under-reporting is especially prevalent in countries where smoking among women is not culturally acceptable.[25] The lack of standardisation across studies in how smoking status was obtained, including how smoking dose and duration were measured is also a major limitation. In addition, there was insufficient data available to examine whether there were sex differences in the impact of age at smoking initiation and smoking duration on the risk of lung cancer. The reference group of non-smokers in our analysis of current smoking was composed of former and never smokers which may inflate the risks of smoking-related lung cancer risk among non-smokers. However, we have also examined former smoking compared with never smoking and demonstrated no appreciable sex differences in the risks of smoking-related lung cancer in this

group, which provides some evidence that the inclusion of former smokers in the reference category is unlikely to have biased the sex difference in our main analysis. We quantified sex differences in the risk of lung cancer associated with smoking based on RRs rather than absolute risks. This might introduce a statistical artefact, in which the generally higher absolute risk for lung cancer in men, and the same risk difference subsequent to smoking in each sex, would translate to a greater RR in women than men. However, our previous meta-analyses on risk factors for cardiovascular diseases demonstrated that sex differences in RRs are not inevitable,[26] despite differences in absolute risks. Compared with absolute risks, RRs are more stable across populations with different background risks, which makes them suitable for meta-analyses. In addition,

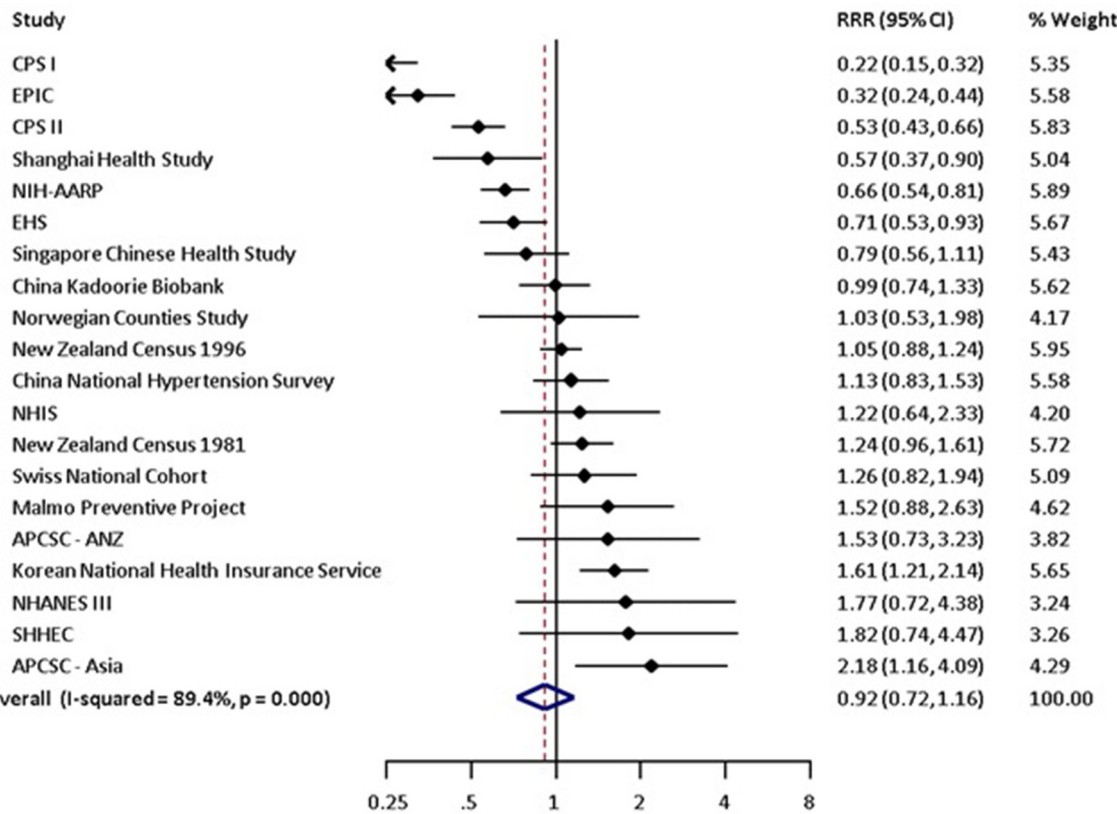

**Figure 3** Multiple-adjusted women-to-men ratio of relative risks (RRR) for incident lung cancer, comparing current smokers to non-smokers. Multiple-adjusted includes anything that adjusted for more than just age. These covariates are listed in table 1. Figures may contain less than 29 studies because we report age-adjusted and multiple-adjusted results separately. Some studies only contributed age-adjusted results whereas others only provided multiple-adjusted results. However, the count of unique studies that contributed to at least one of these analyses is 29. APCSC, Asia Pacific Cohort Studies Collaboration; ARIC, Atherosclerosis Risk in Communities study; CPS, Cancer Prevention Study; EHS, Elderly Health Services; EPIC, European Prospective Investigation into Cancer; JACC, Japan Collaborative Cohort Study; JPHC, Japan Public Health Centre Study; NHANES III, National Health And Nutrition Examination Survey III; NHIS, National Health Interview Survey; NIH-AARP, National Institutes of Health American Association of Retired Persons Diet and Health Study; SHHEC, Scottish Heart Health Extended Cohort Study; TPCS, Three-Prefecture Cohort Study.

RRs are reported much more commonly than absolute risks. In our review, no studies reported adjusted absolute risks, with standard errors, that allow for meta-analyses. We, therefore, believe that use of RRs in the present analysis is appropriate. In addition, while we have aimed to assess study quality using the widely accepted and used NOS, the value and contribution of quality assessment scales such as this to systematic reviews and meta-analyses continues to be debated.[27–29] Finally, there are differences between men and women in histological subtypes of lung cancer. Adenocarcinoma is more common in women and squamous cell carcinoma is more common in men.[30] Smoking is more strongly associated with squamous cell carcinoma than adenocarcinoma.[30] Few studies reported the sex-specific association of smoking with histological subtypes of cancer, which precluded the examination of sex differences in the association of smoking-related lung cancer subtypes and this remains an important limitation of our review.[30] Further studies of the smoking-related risks of lung cancer in women and men are required as the smoking epidemic reaches its full maturity in women. Given the later up-take of smoking in women, studies

which allow sufficient lag time for lung cancer to develop are essential. In addition, reducing under-reporting of smoking in women, using standardised and robust methods for the ascertainment of smoking status and smoking behaviours and more extensive measurement and adjustment for confounders which differ by sex (such as exposure to environmental tobacco smoke) is also important for future work, as well as examination of histological subtypes of lung cancer which was not possible in this review.

In conclusion, this meta-analysis, summarising all available literature to date, shows that the effect of smoking on risk of lung cancer is similar in women and men. However, these data may yet underestimate the true RR of smoking-related lung cancer in women, given later uptake and lower intensity of smoking in women. Although strides have been made in reducing smoking rates particularly in high-income countries, continuing efforts to measure the effects of smoking on disease outcomes are required, as the smoking epidemic has not yet reached its global peak, particularly among women. In addition, tobacco control

**Table 3** Maximally adjusted pooled women to men ratio of relative risks (RRR) for lung cancer associated with smoking, in subgroup analyses

| | N studies | Former versus never | P for interaction* | N studies | Current versus not | P for interaction* |
|---|---|---|---|---|---|---|
| Study region | | | | | | |
| Asia | 40 | 1.07 (0.83 to 1.37) | | 46 | 0.93 (0.72 to 1.20) | |
| Non-Asia | 49 | 0.73 (0.58 to 0.93) | 0.06 | 53 | 0.87 (0.66 to 1.14) | 0.73 |
| Study region | | | | | | |
| Asia | 40 | 1.07 (0.83 to 1.37) | | 46 | 0.93 (0.72 to 1.20) | |
| USA | 5 | 0.60 (0.42 to 0.84) | | 6 | 0.58 (0.37 to 0.91) | |
| Europe | 36 | 0.81 (0.60 to 1.10) | | 37 | 0.99 (0.63 to 1.57) | |
| ANZ | 8 | 1.41 (0.65 to 3.04) | 0.55 | 10 | 1.11 (0.97 to 1.28) | 0.69 |
| Year of study baseline | | | | | | |
| 1985 or before | 23 | 0.79 (0.56 to 1.12) | | 25 | 0.96 (0.66 to 1.40) | |
| After 1985 | 66 | 0.92 (0.73 to 1.16) | 0.43 | 74 | 0.85 (0.68 to 1.06) | 0.66 |
| Women-to-men smoking prevalence | | | | | | |
| >67% lower in women | 39 | 1.16 (0.83 to 1.62) | | 43 | 0.96 (0.72 to 1.28) | |
| 33%–67% lower in women | 19 | 0.72 (0.51 to 1.03) | | 20 | 0.75 (0.50 to 1.14) | |
| 0%–33% lower in women | 28 | 0.78 (0.59 to 1.03) | 0.26 | 29 | 0.97 (0.59 to 1.58) | 0.90 |
| Women-to-men lung cancer rate | | | | | | |
| ≥50% lower in women | 80 | 0.85 (0.65 to 1.10) | | 83 | 0.85 (0.62 to 1.17) | |
| 0%–50% lower in women | 6 | 0.83 (0.61 to 1.13) | 0.79 | 9 | 0.94 (0.68 to 1.31) | 0.64 |
| Study endpoint | | | | | | |
| Fatal lung cancer only | 57 | 0.94 (0.68 to 1.29) | | 67 | 0.97 (0.77 to 1.21) | |
| Fatal and non-fatal lung cancer | 32 | 0.80 (0.64 to 1.01) | 0.57 | 32 | 0.72 (0.48 to 1.06) | 0.21 |
| Duration of follow-up | | | | | | |
| ≤10 years | 48 | 0.90 (0.60 to 1.35) | | 53 | 0.91 (0.68 to 1.24) | |
| >10 years | 41 | 0.85 (0.71 to 1.05) | 0.92 | 46 | 0.87 (0.67 to 1.12) | 0.83 |
| Study quality | | | | | | |
| ≤6 points | 58 | 0.85 (0.64 to 1.13) | | 61 | 0.84 (0.53 to 1.12) | |
| >6 points | 31 | 0.89 (0.66 to 1.20) | 0.81 | 38 | 0.93 (0.72 to 1.20) | 0.69 |

Random-effects meta-analyses were used for all subgroup analyses and differences between subgroups were examined using meta-regression.
*P for interaction assessed using meta-regression.
ANZ, Australia and New Zealand.

programmes that dissuade both sexes from smoking but which also encourage individuals to quit remain a priority.

**Author affiliations**
[1]MRC Integrative Epidemiology Unit at the University of Bristol, Bristol Medical School, University of Bristol, Bristol, UK
[2]Population Health Sciences, Bristol Medical School, University of Bristol, Bristol, UK
[3]UK Centre for Tobacco and Alcohol Studies, School of Experimental Psychology, University of Bristol, Bristol, UK
[4]Department of Psychology, University of Bath, Bath, UK
[5]College of Science, Health and Engineering, La Trobe University, Melbourne, Australia
[6]The George Institute for Global Health, University of New South Wales, Sydney, New South Wales, Australia
[7]Olivia Newton-John Cancer and Wellness Centre, Austin Health and Olivia Newton-John Cancer Research Institute, Heidelberg, Victoria, Australia
[8]The George Institute for Global Health, University of Oxford, Oxford, UK
[9]Department of Epidemiology, John Hopkins University, Baltimore, Maryland, USA

**Contributors** LMOK, GT, SAEP, RRH and MW designed the study. LMOK and GT performed systematic searches, retrieved literature and performed data extraction. SAEP performed the analysis of the data. LMOK and GT wrote the first draft of the article. RRH, MW and PM contributed important intellectual content and critical expertise and revisions to the manuscript.

**Funding** The MRC Integrative Epidemiology Unit at the University of Bristol is supported by the University of Bristol and the Medical Research Council [MC_UU_12013/2, MC_UU_12013/3, MC_UU_12013/4, MC_UU_12013/6, MC_UU_12013/9]. LMOK is supported by a UK Medical Research Council Population Health Scientist fellowship (MR/M014509/1). GT is funded by a Cancer Research UK Postdoctoral Fellowship (C56067/A21330). SAEP is supported by a UK Medical Research Council Skills Development Fellowship (MR/P014550/1).

**Competing interests** None declared.

**Patient consent** Not required.

**Provenance and peer review** Not commissioned; externally peer reviewed.

**Data sharing statement** Technical appendix, statistical code and dataset (of published data only) available from authors on request.

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
