## [Reviewer comments · BMJ Open]

ARTICLE DETAILS

TITLE (PROVISIONAL)	Smoking as a risk factor for lung cancer in women and men: a systematic review and meta-analysis
AUTHORS	O'Keeffe, Linda; Taylor, Gemma; Huxley, Rachel; Mitchell, Paul; Woodward, Mark; Peters, Sanne

VERSION 1 – REVIEW

REVIEWER	Joshua Muscat Penn State University, USA
REVIEW RETURNED	29-Jan-2018

GENERAL COMMENTS	This is a well performed MA on a topic that has been of interest for several years. It makes a nice contribution to the literature. There are a few issues that would be helpful to clarify. 1. The abstract is a little misleading in giving the impression that the MA was based on 99 studies. It should include the qualifier that 26 studies were eventually used for quantitative MA.2. The introduction needs elaboration. The assumption is that women may be more susceptible to smoke. That is a host factor. But it may be that historically more women smokers had a greater history of smoking filtered vs unfiltered cigarettes, or cigarettes with lower tar yields and therefore the expectation might be that they would have lower risks.3. The discussion section also needs further discussion on the histologic-specific risks, which was not available in the current study. Women have traditionally more likely to get adenocarcinoma, which is less related to smoking. The fact that AC is less related to smoking but more common among women makes it difficult to make any firm conclusions about the findings. Were histologist-specific risks available? If not, the authors should state that. If they were in a subset of studies, it would be valuable to add this to the paper.4. While it is stated that women are more exposed to ETS in the discussion, it should be stated more explicitly that this might result in an underestimate of the pooled RR for women.5. The findings should be compared to the published MA for case-control studies. Do they differ and why?
---

REVIEWER	Yunxian Yu Zhejiang University, China
REVIEW RETURNED	15-Feb-2018

GENERAL COMMENTS	This meta-analysis includes 26 cohort studies. The results from cohort studies should be more believable than those from case-control studies. This meta-analysis found that there is not difference in the effect of smoking on risk of lung cancer between two genders. It is different from the previous meta-analysis. However, the following suggestions and comments need be considered:  1. The study quality did not be described in the manuscript. 2. The sub-analysis of the following factors should be considered: the time period of follow-up in the cohorts (<10,>=10 years), duration of smoking, study quality. 3. The homogeneity is precondition of merging the results from different studies. Generally, while the heterogeneity (I²) is greater than 75%, the pooled effect may be unstable. However, despite of several I²>75%, the authors still provided the pooled effects in the current manuscript without more explanation. Based on those results, the conclusion may not be reasonable. 4. The recent meta-analysis about the literature titled as "gender susceptibility for cigarette smoking-attributable lung cancer" published in 2014 LUNG CANCER was not mentioned by authors. The different findings between the literature and current manuscript should be discussed in the manuscript. 5. The gender difference of smoking prevalence, especially among the population with high smoking prevalence of men and low smoking prevalence of women, results in the higher passive smoking for women without smoking. Therefore, it may distort the gender difference of association between smoking and lung cancer. Additionally, the smoking level in women may be lower than that in men, due to social environments and culture background. These reasons may be considered in discussion section.
--

REVIEWER	Marzieh Nojomi Preventive medicine and public health research center. Iran University of medical sciences
REVIEW RETURNED	07-Mar-2018

GENERAL COMMENTS	This is a well designed systematic review. Just I suggest to review by a statistician
---

REVIEWER	Tao Chen Liverpool School of Tropical medicine, UK
REVIEW RETURNED	03-Apr-2018

GENERAL COMMENTS	In this study, the authors are trying to explore the sex-specific association between smoking and lung cancer through several cohort studies. the study has advantages in terms of large number of studies, sample size and cases of lung cancer. Several concerns are needed to be addressed.  1, the non-smokers in this study are composed of former and never smokers. This may dilute the effect size between smoker and never smoker. 2, the variation of RR across different studies are quite large from 1.38 to 29.40. This could be explained by the heterogeneities among studies, which are stated in the limitation. However, is it possible to present some more data to test the impact of deleting the study one by one?
---

	3, please specify which variable is being used as the random effect in the mixed model. 4, please add the number of studies in each subgroup for better digesting the results. 5, it is better to add some sentences to highlight problems of standardized information across studies in strengths and limitations
--	---

REVIEWER	Dr Muhammad Riaz Department of Health Sciences, University of Leicester United Kingdom
REVIEW RETURNED	06-Apr-2018

GENERAL COMMENTS	Review for bmjopen-2018-021611: Smoking as a risk factor for lung cancer in women and men: a systematic review and meta-analysis This paper is addressing an important research question of whether gender specific differences exist in the risk of smoking related lung cancer by reviewing (23/34=?) articles /studies (number of cohorts 99=?) and including 20 in pooling the effect estimates. The authors have followed a better approach in conducting systematic review and meta-analysis to answer the research question. The authors are commended for a huge amount of work which they have put into accomplishing this task, and producing a well-written paper. While this review and meta-analysis has the potential to make a valuable contribution to the field and I think would be of interest to the readers in the field, I would like to comment and suggest a major revision. I believe the changes I suggest will improve the manuscript. Comments to the authors: Abstract: Results: Abstract must include briefly the statistical methods used for the analysis. 1. "Data from 99 cohort studies" sounds like 99 results would have been pooled together, but it is not. This need clarity on the number of results pooled. Strengths and Limitations: 2. This should be deleted from abstract, as it is included in the discussion section in more detail. Deleting this will provide room for adding further results (possible results for sub-groups analyses) to the abstract. Introduction section: 3. In the last paragraph of introduction on p.4, the authors need to explicitly present a strong case for the need to conduct an updated and extended systematic review for identifying the gender specific difference in the risk of lung cancers caused by smoking since that was reported by (reference 4: Lee et al 2012). References are required for "An important a priori consideration is the substantial sex difference in the maturity of the smoking epidemic with men being at a more advanced stage than women in most parts of the world" "This would be expected to translate into lower relative risk estimates for lung cancer in women than in men. Hence, the null
--

	hypothesis that smoking confers the same lung cancer hazard in both women in men, would be met if the ratio of the relative risks for lung cancer (women:men) was less than unity (reflecting a greater hazard in men than women). However, if the ratio of the relative risks was found to be unity (or higher) then this would suggest greater hazard associated with tobacco exposure in women than in men.” Here they could further discuss a few major limitations of the previous work and it would be better to say that this kind of work was not done previously. Methods section: Generally speaking the review is conducted and presented well. The paper describes clearly the review process, including use of MOOSE guidelines, searching process, inclusion/exclusion criteria, statistical analysis, assessment of study quality, and resolution of conflicts between reviewers. Search strategy: 4. p.5: I suggest adding a clear description of the search criteria in plain text in the method section, the computer search criteria (MeSH terms), presented in Appendix 2 is not easy for a common reader to follow. 5. p.5: Although I understand that age adjusted effect estimates are better in this kind of review, I think it would have been better if study with unadjusted effect estimates had been included. If there are many studies, which report the un-adjusted effect estimates, the results could have been pooled separately as a sensitivity analysis. If the effect estimates are not readily available, it could be computed from the data published in the paper. This would also increase the statistical power for the analysis. 6. It is not clear what modified version of the Newcastle-Ottawa Quality assessment scale (NOQAS) was used, a reference is required on p.5 if it is previously used. If they have modified the NOQAS for this study, it should be explained on p.5. Data extraction? 7. It would have been better to have a sub-section of “Data extraction” in the methods section and explain what kind of data/results were extracted. Also, the last sentence in the search strategy could be moved to “data extraction” section. Statistical analysis 8. I suggest changing this sub-heading of this sub-section to Meta-analysis. 9. It is mentioned that RRs were extracted, I am interested to know if there were any studies which have reported the effect estimates in the form of other statistics, (e.g., ORs, HR etc), If so, a detail in the statistical analysis is required whether different measures (statistics) were pooled together. 10. p.6: Should the “difference” be “difference [Log(RR) for women-Log(RR) for men] ? 11. p.6: “The standard error of the log RRR was calculated as the
--	--

square root of the sum of the variance of the two sex-specific log RRs for each study.”

Should this be: square root of $(\text{var}(\text{women log(RR)/n1 of studies}) + (\text{var}(\text{women log(RR)/n2 of studies}))$?. If so, please make the correction.

Results section:

12. p7. Clarity is required on the number of studies included and the number of cohorts. 23 articles from 25 studies? where adding the three further (published/unpublished studies=?) n=28 studies? The number of cohorts=99, how? A clear explanation is required in the text.

13. As suggested above, if the authors decide to add the studies reporting unadjusted effect estimates. They may consider computing the measure of association for the studies where it is not reported but relevant data is available in the published paper or authors of the relevant studies could be contacted for data if possible.

14. In the figures (forest plots and funnel plots), I can track only a maximum of 20 results pooled together, but in the results section 23 articles from 25 studies or even possibly 28, (while there are 34 records of studies in table 1). The number of studies and number results included in the pooled analysis should be consistent otherwise an explanation is needed in the text as to why there is a discrepancy.

15. Table 1 and forest plots are confusing in terms of referring to a study name (with more than one references) rather results from the included published papers. A standard way of citation (Harvard reference, 1st author's surname et al., year) would have been better. It does not matter if numerical numbers for references have been used in the main body. I am not quite sure why a study name is more appropriate than a reference (citation) to the published paper from which the measure of association has been extracted. Also, it would be better to arrange studies in the table by the alphabetical order of the authors names. These comments also apply to all forest plots in the figures. If for some of the studies, the authors had to use more than one articles, I think it would have been better to add the main article, which provide the measure of association in the tables and figures, and add a footnote to the tables that additional data is extracted from other articles with reference number X.

16. In table 1, a column on methods used in the study for assessing the association (e.g., main aim, study design, and statistical analysis, measure of association) would have been better. I suggest moving this table 1 to an eTable and giving its place in the main paper to a modified eTable2 as I suggest in the point below.

17. eTable 2, I suggest moving this table to the main paper and name it Table1 and add the effect estimates of each original study above the pooled estimates. Also, add explanation in the footnote to the table wherever it is required for various things, such as explaining what is meant by multiple adjustment, maximum adjustment etc.

18. eFigure 3: Instead of funnel plot a contour funnel plot will be

more useful. It allows the statistical significance of study estimates to be considered, see (Peters et al., 2008) for more detail.

19. Publication bias: no results from “Begg's test” OR “Egger's test” are included in the results section, at least a p-value could have been provided in the results section and also added under the funnel plot.

20. While I appreciate the author's efforts on quality assessment, I would like to see some acknowledgment of the limitations of NOQAS in the discussion section. A couple of useful references:

Sanderson S, Tatt ID, Higgins JPT. Tools for assessing quality and susceptibility to bias in observational studies in epidemiology: a systematic review and annotated bibliography. *Int J Epidemiol* 2007;36(3):666-676. doi:10.1093/ije/dym018

Katikireddi SV, Egan M, Petticrew M. How do systematic reviews incorporate risk of bias assessments into the synthesis of evidence? A methodological study. *J Epidemiol Community Health* 2015;69(2):189-195. doi:10.1136/jech-2014-204711

Viswanathan M, Berkman ND, Dryden DM, Hartling L. Assessing risk of bias and confounding in observational studies of interventions or exposures: further development of the RTI item bank. 2013.

Available from:<http://www.effectivehealthcare.ahrq.gov/search-for-guides-reviews-and-reports/?pageaction=displayproduct&productid=1612> NOTE: Table 1 is especially helpful.

21. I suggest an extended table for the NOQAS having the following information:

Selection:

REC: Representativeness of exposed cohort (i.e., Is the exposed cohort ‘truly’ or ‘somewhat’ representative of exposed group)

SNEC: Selection of the non-exposed cohort? (i.e., Is the non-exposed cohort drawn from the same community as the exposed cohort?).

AE: Ascertainment of exposure (i.e., Is ascertainment based on either medical records or a structured interview?).

DONPS: Demonstration that outcome of interest was not present at start of the study.

Comparability:

Design: Study controls for the most important factor (i.e., socio-economic and demographic status for cohort studies).

Analyses: Study adjusts results for additional potential confounders

Outcome:

Assessment: Assessment of outcome (i.e., Is self-reported lung cancer validated by a medical test).

LFU: Was follow-up long enough for outcome to occur?

AFUC: Adequacy of follow-up of cohorts (i.e., was % loss to follow-up low (<10%) or with no observable bias or with an appropriate assumption made to the outcome of lung cancer in those lost to follow-up?).

22. Further to the quality assessment, there is a lack of presenting the results. I think based on NOQAS it should be clearly explained, how the authors declared a study to be of a good quality? In addition, as table 1 describes the characteristics of the studies and is published in the main paper, it will be helpful to add the final NOQAS to this table. I wondered, whether the women-to-men RRR of lung cancer would differ by the quality of the study, possibly NOQAS \geq 7 to be classed as of high quality.

23. On p.7, the ranges of prevalence of smoking and smoking cessation for women and men have been reported. It will be helpful if the prevalence can also be pooled separately for each sex and combined. This is important as the author argue that smoking has not reached its maturity in women. The pooled provenances of smoking could also be compared between the two sexes.

24. Explaining the context of maturity of smoking in women, studies have shown that there was a greater disparity of smoking between women and men in oldest population, but this has decreased substantially in the youngest population (see peters et al 2014). In western population, “the women-to-men ever-smoking ratio ranged from 0.57 in the oldest to 0.87 in the youngest birth cohort”. If the trend continues, one may assume that there will be no difference. The results should be explained in the light of these results. Also, as they have got smoking provenance data for both sexes, they may be able to adjust the results for the interaction of smoking prevalence and sex, this may explain the difference if it exists.

25. There is a substantial amount of between-study heterogeneity, therefore meta-regression is inevitable, it is not explained why the author did not conduct meta-regression to explore the sources heterogeneity using the study level covariates/moderators to understand these diverse findings. I strongly recommend the use of meta-regression; this might be helpful in explaining the sex differences.

Also, as reported for some of the studies, subject level data is also available, if it is believed that there might be sex-related differences, I wondered if subject level data has been explored after the meta-analysis.

26. I think text on p.8 in the results section and related Table 2 need revision; it is difficult to understand what it means, also in the text below RRR and table 2 heading RR:

“There was no evidence for a difference in the RRR according to the women-to-men ratio of current smokers ($P=0.90$) or the women-to-men ratio of lung cancer incidence in the studies ($P=0.64$) (**Table 2**).”

The statistical analysis section does not explain clearly, what statistical methods was used for these analyses (results in Table 2). An explanation in the footnote under table 2 should be added.

27. It is not clear in table 1 and text, how smoking has been assessed in the original studies. Some data on smoking (self-reported, or biochemically validated) could have been added to table 1. Similarly, assessment of the outcome of lung-cancer (self-reported, or diagnosed/ascertained by a medical test) for each study need to be added to table 1.

Discussion:

28. In the 1st paragraph of discussion the results are summarized well, and the authors further explanation for no difference among women and men might be reasonable. This might need some published studies support (reference) to confirm that smoking is not matured among women. Currently, it seems like the explanation is based on speculation.

On the basis of their findings, they may recommend further research in a well-designed study, where it could be typically assessed

	whether any gender-specific difference in the outcome of lung cancer exist among smokers. 29. As the study has extracted data from studies published globally, it might be useful if the difference of smoking related lung-cancer between women and men is explored among the high and low-income countries (if possible). 30. p.10, 2nd paragraph “The relatively small relative risk estimates observed in this study are likely to partly reflect the heterogeneity in study populations in terms of baseline year of study, population age, prevalence of smoking, and smoking duration and intensity”: The author need to clarify, does “small relative risk estimates” refer to separate risk (RR) for women and men or women to men ratio of RR. This explanation of heterogeneity is not consistent with the results and statistical methods. The authors need to explore heterogeneity using Meta-regression. 31. Reference 1: “Institute for Health Metrics and Evaluation. Global Burden of Disease 2015. 2015. http://vizhub.healthdata.org/gbd-compare/#” need to be archived online.
--	---

VERSION 1 – AUTHOR RESPONSE

Reviewer: 1

Reviewer Name: Joshua Muscat

Institution and Country: Penn State University, USA

Please state any competing interests or state ‘None declared’: None

Comments to the Author

Overall comment: This is a well performed MA on a topic that has been of interest for several years. It makes a nice contribution to the literature. There are a few issues that would be helpful to clarify.

Concern #3: The abstract is a little misleading in giving the impression that the MA was based on 99 studies. It should include the qualifier that 26 studies were eventually used for quantitative MA.

Response: We understand that this may be misleading. We did include 29 studies (26 from the published literature and 3 studies for which we had individual participant data) but these represented data from 99 cohorts overall. This is now clarified on page 2, lines 19-20:” *Data from 29 studies representing 99 cohorts, seven million individuals and over 50,000 incident lung cancer cases were included.*”

Concern #4: The introduction needs elaboration. The assumption is that women may be more susceptible to smoke. That is a host factor. But it may be that historically more women smokers had a greater history of smoking filtered vs unfiltered cigarettes, or cigarettes with lower tar yields and therefore the expectation might be that they would have lower risks.

Response: We agree that differences in smoking behaviour may be a factor and there is evidence that men more often smoke cigars and pipes and have smoking behaviours which are consistent with greater risk. This may, as the reviewer points out, lead to lower risks in women and this point is alluded to now on page 4, lines 16-20 of the introduction as follows: *“In addition, smoking behaviour varies significantly by sex. For example, compared with women, men smoke more cigars and pipes,(5) take puffs of longer duration, and leave shorter butts,(6) which each could potentially predispose them to greater risks of smoking-related lung cancer.”*

Concern #5: The discussion section also needs further discussion on the histologic-specific risks, which was not available in the current study. Women have traditionally more likely to get adenocarcinoma, which is less related to smoking. The fact that AC is less related to smoking but more common among women makes it difficult to make any firm conclusions about the findings. Were histologist-specific risks available? If not, the authors should state that. If they were in a subset of studies, it would be valuable to add this to the paper.

Response: We agree with the reviewer that histologic-specific risks are important, but – as suggested – such data are sparse. We have thus addressed differences in histological types as follows on page 15, lines 15-28 as follows: *“There are differences between men and women in histological subtypes of lung cancer. Adenocarcinoma is more common in women and squamous cell carcinoma is more common in men. (4) Smoking is more strongly associated with squamous cell carcinoma than adenocarcinoma. (4) Few studies reported the sex-specific association of smoking with histological subtypes of cancer, which precluded the examination of sex differences in the association of smoking-related lung cancer subtypes and this remains an important limitation of our review. (4)”*

Concern #6: While it is stated that women are more exposed to ETS in the discussion, it should be stated more explicitly that this might result in an underestimate of the pooled RR for women.

Response: We agree with the reviewer that exposure to ETS is an important one in women. We have highlighted this as follows on page 13, lines 22-30: *“The possibility of greater exposure to environmental tobacco smoke and other environmental carcinogens in women compared with men could have resulted in a greater underestimation of the association between smoking and lung cancer in women than men. This, in turn, could have impacted the sex difference in risk of smoking-related lung cancer reported in this study. “*

Concern #7: The findings should be compared to the published MA for case-control studies. Do they differ and why?

Response: We have now addressed this as follows in the discussion on page 13, lines 36-43: *“Differences between case-control and cohort studies may also explain why a previous meta-analysis of case-control studies (which included only 3 cohort studies) showed higher relative risk of lung cancer in men compared with women.(2)”*

Reviewer: 2

Reviewer Name: Yunxian Yu

Institution and Country: Zhejiang University, China

Please state any competing interests or state ‘None declared’: None declared

Comments to the Author

Overall comment: This meta-analysis includes 26 cohort studies. The results from cohort studies should be more believable than those from case-control studies. This meta-analysis found that there is not difference in the effect of smoking on risk of lung cancer between two genders. It is different from the previous meta-analysis. However, the following suggestions and comments need be considered:

Concern #8: The study quality did not be described in the manuscript.

Response: We have now described this in the manuscript at follows on page 9, lines 28-30: *“Of 29 studies, four studies had a quality score of 5 out of 9, 9 studies had a score of 6, 12 studies had a score of 7 and 4 studies with a score of 8 (eTable 1)”* A detailed table of study quality is also provided in the Supplementary Material.

Concern #9: The sub-analysis of the following factors should be considered: the time period of follow-up in the cohorts (<10,>=10 years), duration of smoking, study quality.

Response: We now provide a subgroup analysis by duration of follow-up and study quality, the details of which are contained in Table 3. Sex-specific data on duration of smoking were not widely reported and therefore could not be included in the subgroup analyses.

Concern #10: The homogeneity is precondition of merging the results from different studies. Generally, while the heterogeneity (I²) is greater than 75%, the pooled effect may be unstable. However, despite of several I²>75%, the authors still provided the pooled effects in the current manuscript without more explanation. Based on those results, the conclusion may not be reasonable.

Response: While we agree that homogeneity assumptions are important, they can be extremely unlikely to be satisfied, given the differences in covariates, bias, and exposure variables across the studies (7). Furthermore, we only draw a conclusion about the general (average) case using random effects meta-analysis, which produces confidence intervals that account for study-to-study homogeneity. In addition, we have examined potential sources of heterogeneity in our subgroup analyses and also acknowledged the issue of heterogeneity in our discussion.

Concern #11: The recent meta-analysis about the literature titled as “gender susceptibility for cigarette smoking-attributable lung cancer” published in 2014 LUNG CANCER was not mentioned by authors. The different findings between the literature and current manuscript should be discussed in the manuscript.

Response: Thank-you for pointing us to this important paper. As also requested by the associate editor and reviewer 1, we have now addressed this in the discussion page 13, lines 34-43.

Furthermore, we raise the comparison and the differences in findings again briefly in the discussion as follows on page 13, lines 36-43: *“Differences between case-control and cohort studies may also explain why a previous meta-analysis of case-control studies (which included only 3 cohort studies) showed higher relative risk of lung cancer in men compared with women.(2)”*

Concern #12: The gender difference of smoking prevalence, especially among the population with high smoking prevalence of men and low smoking prevalence of women, results in the higher passive smoking for women without smoking. Therefore, it may distort the gender difference of association between smoking and lung cancer. Additionally, the smoking level in women may be lower than that in men, due to social environments and culture background. These reasons may be considered in discussion section.

Response: The issue of greater exposure to environmental tobacco smoke is one also raised by reviewer 1. We address this as follows on page 13, lines 22-30: *“The possibility of greater exposure to environmental tobacco smoke and other environmental carcinogens in women compared with men could have resulted in underestimation of the association between smoking and lung cancer to a greater degree in women than men. This, in turn, could have impacted the sex difference in risk of smoking-related lung cancer reported in this study. “*

The variation in the women to men smoking prevalence across different world regions and its potential impact on our findings is addressed by our subgroup analyses by categories of smoking prevalence. On page 13, line 49 to page 14, line 5 we report: *“Our results were consistent across regions and irrespective of the women-to-men smoking ratio, suggesting that underestimation of the association of smoking with lung cancer in women due to sex differences in smoking prevalence and underreporting of smoking is unlikely. This is especially relevant for parts of Asia where the prevalence of smoking in women is typically <10% and where smoking among women remains relatively socially unacceptable.”*

Furthermore, we raise the issue of underreporting of smoking in women in regions where smoking is culturally unacceptable as follows on page 14, lines 17-22: *“Assessment of smoking status differed across studies and was generally self-reported, which may have introduced measurement error.(8) Notably, compared with men, women typically are more likely to underreport smoking status, and underreporting is especially prevalent in countries where female smoking is not culturally acceptable.(9)”*

Reviewer: 3

Reviewer Name: Marzieh Nojomi

Institution and Country: Preventive medicine and public health research center. Iran University of medical sciences

Please state any competing interests or state ‘None declared’: None

Comments to the Author

Overall comment: This is a well-designed systematic review. Just I suggest reviewing by a statistician

Reviewer: 4

Reviewer Name: Tao Chen

Institution and Country: Liverpool School of Tropical medicine, UK

Please state any competing interests or state 'None declared': None declared

Comments to the Author

Overall comment: In this study, the authors are trying to explore the sex-specific association between smoking and lung cancer through several cohort studies. The study has advantages in terms of large number of studies, sample size and cases of lung cancer. Several concerns are needed to be addressed.

Concern #13: The non-smokers in this study are composed of former and never smokers. This may dilute the effect size between smoker and never smoker.

Response: We agree with the reviewer. In our limitations section we report on page 14, lines 32-45: *"The reference group of non-smokers in our analysis of current smoking was composed of former and never smokers which may inflate the risks of smoking-related lung cancer risk among non-smokers. However, we have also examined former smoking compared with never smoking and demonstrated no appreciable sex differences in the risks of smoking-related lung cancer in this group, which provides some evidence that the inclusion of former smokers in the reference category is unlikely to have biased the sex difference in our main analysis."*

Concern #14: The variation of RR across different studies are quite large from 1.38 to 29.40. This could be explained by the heterogeneities among studies, which are stated in the limitation. However, is it possible to present some more data to test the impact of deleting the study one by one?

Response: We agree that the variation in the size of the RRs across studies could be explained by between-study heterogeneity. However, the range across studies for our primary endpoint, i.e. the women-to-men ratios of RRs, was much smaller (from 0.22 to 2.18). We used random-effects meta-analyses to account for heterogeneity between studies. As a result, the impact of each study on the pooled results was relatively similar, compared with a simple fixed effect meta-analysis, across studies. This can be seen by the weight that each study had in our analyses, which ranged from 3.24% for the least precise study (NHANES III) to 5.95% for the most precise study (New Zealand Census 1996). We have addressed the issue of heterogeneity in our limitations section.

Concern #15: Please specify which variable is being used as the random effect in the mixed model.

Response: The random effect is the study. We used meta-analyses with random-effects, which allows the effect of smoking on the risk of lung cancer to vary from study to study. We have added this to the revised manuscript. See page 7, lines 26 - 32.

Concern #16: Please add the number of studies in each subgroup for better digesting the results.

Response: We have added the number of cohorts to Table 2. This table is now labelled Table 3 due to the placing of eTable 2 into the main text, as requested by reviewer #5.

Concern #17: it is better to add some sentences to highlight problems of standardized information across studies in strengths and limitations.

Response: This issue is mentioned on page 11, lines 26 - 28 as follows: *“The lack of standardisation across studies in how smoking status was obtained, including how smoking dose and duration were measured is also a limitation.”*

Reviewer: 5

Reviewer Name: Dr Muhammad Riaz

Institution and Country: Department of Health Science: University of Leicester, United Kingdom

Please state any competing interests or state ‘None declared’: None

Comments to the author

Overall comment: This paper is addressing an important research question of whether gender specific differences exist in the risk of smoking related lung cancer.

The review and meta-analysis has the potential to make a valuable contribution to the field and I think would be of interest to the readers in the field, but I suggest a major revision particularly to the statistical part of the paper. I have changes in the attached document and I believe this will improve the manuscript.

This paper is addressing an important research question of whether gender specific differences exist in the risk of smoking related lung cancer by reviewing (23/34=?) articles /studies (number of cohorts 99=?) and including 20 in pooling the effect estimates. The authors have followed a better approach in conducting systematic review and meta-analysis to answer the research question. The authors are commended for a huge amount of work which they have put into accomplishing this task and producing a well-written paper. While this review and meta-analysis has the potential to make a valuable contribution to the field and I think would be of interest to the readers in the field, I would like to comment and suggest a major revision. I believe the changes I suggest will improve the manuscript.

Concern #18: Abstract: Results: Abstract must include briefly the statistical methods used for the analysis.

Response: This has now been added to the abstract on page 2, lines 20 - 21: *“The sex-specific RRs and their ratio comparing women with men were pooled using random-effects meta-analysis with inverse-variance weighting.”*

Concern #19: Abstract: “Data from 99 cohort studies” sounds like 99 results would have been pooled together, but it is not. This need clarity on the number of results pooled.

Response: We understand that this may be confusing. We did indeed include 29 studies but these represented data from 99 cohorts overall. This is now clarified on page 2, lines 19 to 20: *“Data from 29 studies representing 99 cohorts, seven million individuals and over 50,000 incident lung cancer cases were included.”*

Concern #20: Abstract: Strengths and Limitations: This should be deleted from abstract, as it is included in the discussion section in more detail. Deleting this will provide room for adding further results (possible results for subgroups analyses) to the abstract.

Response: A separate strengths and limitations section is a requirement for publication in BMJ Open. We have now separated this from the abstract.

Concern #21: Introduction section: In the last paragraph of introduction on p.4, the authors need to explicitly present a strong case for the need to conduct an updated and extended systematic review for identifying the gender specific difference in the risk of lung cancers caused by smoking since that was reported by (reference 4: Lee et al 2012).

Response: The case for an updated systematic review and meta-analysis is now described as on page 4, line 51 to page 5, line 11.

Concern #22: References are required for “An important a priori consideration is the substantial sex difference in the maturity of the smoking epidemic with men being at a more advanced stage than women in most parts of the world” “This would be expected to translate into lower relative risk estimates for lung cancer in women than in men. Hence, the null hypothesis that smoking confers the same lung cancer hazard in both women in men, would be met if the ratio of the relative risks for lung cancer (women: men) was less than unity (reflecting a greater hazard in men than women). However, if the ratio of the relative risks was found to be unity (or higher) then this would suggest greater hazard associated with tobacco exposure in women than in men.” Here they could further discuss a few major limitations of the previous work and it would be better to say that this kind of work was not done previously.

Response: We have now added a reference for this as requested. We believe the limitations of previous reviews and what our study adds are addressed in the previous response and we also mention these limitations in the discussion section on page 13 lines 34 - 49 as follows: *“Our study has several strengths including restriction to cohort studies which provide more robust evidence of the associations compared with case-control studies. Differences between case-control and cohort studies may also explain why a previous meta-analysis of case-control studies (which included only 3 cohort studies) showed higher relative risk of lung cancer in men compared with women.(2) Other strengths to our study include an update of findings to include studies published after 1999,(3) with supplementation of published literature by individual participant data from three established population databases. We have also performed a range of pre-specified sensitivity analyses and several subgroup analyses which were not performed in previous meta-analyses.”*

Concern #23: Methods section: Generally speaking the review is conducted and presented well. The paper describes clearly the review process, including use of MOOSE guidelines, searching process, inclusion/exclusion criteria, statistical analysis, assessment of study quality, and resolution of conflicts between reviewers.

Concern #24: Search strategy: 4. p.5: I suggest adding a clear description of the search criteria in plain text in the method section, the computer search criteria (MeSH terms), presented in Appendix 2 is not easy for a common reader to follow. 5. p.5:

Response: This has now been added on page 6, lines 13-15 as follows: *“The computer-based searches combined MeSH and free-text terms related to “tobacco/smoking”, “cancer”, “sex” and “cohort studies”*

Concern #25: Although I understand that age adjusted effect estimates are better in this kind of review, I think it would have been better if study with unadjusted effect estimates had been included. If there are many studies, which report the unadjusted effect estimates, the results could have been pooled separately as a sensitivity analysis. If the effect estimates are not readily available, it could be computed from the data published in the paper. This would also increase the statistical power for the analysis.

Response: Age is a key confounder in many epidemiological associations, including that between smoking and lung cancer. By design, and consistent with our series of meta-analyses on sex differences in the effects of risk factors on cardiovascular outcomes, we therefore required each RR to at least be adjusted for age. We believe that this is appropriate and ensures that the included studies fulfil this basic quality standard. Moreover, given the large number of studies that met our criteria for inclusion, only including higher-quality estimates gave us sufficient statistical power to address our objectives.

Concern #26: It is not clear what modified version of the Newcastle-Ottawa Quality assessment scale (NOQAS) was used, a reference is required on p.5 if it is previously used. If they have modified the NOQAS for this study, it should be explained on p.5.

Response: A reference to this scale has now been added. We have added the scale to the supplement of the paper which includes details of the specific criteria we used to assess quality.

Concern #27: Data extraction 7. It would have been better to have a sub-section of “Data extraction” in the methods section and explain what kind of data/results were extracted. Also, the last sentence in the search strategy could be moved to “data extraction” section. Statistical analysis 8. I suggest changing this sub-heading of this sub-section to Meta-analysis. 9.

Response: We have added a section called data extraction (see page 6, lines 30-43). We have not moved the last sentence to the search strategy under “data extraction” as this sentence refers to screening. We have relabelled our statistical analysis section as requested.

Concern #28: It is mentioned that RRs were extracted, I am interested to know if there were any studies which have reported the effect estimates in the form of other statistics, (e.g., ORs, HR etc.), If so, a detail in the statistical analysis is required whether different measures (statistics) were pooled together. 10.

Response: In the methods section, we state that ‘*Observational cohort studies were included if they reported sex-specific relative risks (RRs), or equivalent, on the relationship between smoking and lung cancer.*’ Although we did not track the number of occasions other effect measures were used, most studies would have reported hazard ratios, which is the preferred outcome measure in most studies dealing with survival data.

Concern #29: p.6: Should the “difference” be “difference [Log(RR) for women-Log(RR) for men] ? 11. p.6: “The standard error of the log RRR was calculated as the square root of the sum of the variance of the two sex-specific log RRs for each study.” Should this be: square root of ($[\text{var}(\text{women log(RR)/n1 of studies]} + (\text{var}(\text{women log(RR)/n2}$

Response: As stated in the methods section, we “*obtained the natural log of the sex-specific RRs and calculated the differences*”. This is equivalent to the log RRR. We did this for each study separately. Hence, the standard error for the log RRR was the square root of the sum of the women- and men specific log RRs.

Concern #30: Results section: 12. p7. Clarity is required on the number of studies included and the number of cohorts. 23 articles from 25 studies? where adding the three further (published/unpublished studies=?) n=28 studies? The number of cohorts=99, how? A clear explanation is required in the text.

Response: We included 29 unique studies representing 99 cohorts in total. The 29 studies are represented in Table 1, with a new column for the number of cohorts per study. Other than Table 1, all other figures and tables contain less than 29 studies because we report age- and multiple-adjusted results separately. Some studies only contributed age-adjusted results whereas others only provided multiple-adjusted results. However, the count of unique studies that contributed to at least one of these analyses is 29. When less than 29 studies are presented, we have now provided footnotes explaining this clearly (see legends of Figure 2 & 3 on page 17).

Concern #31: 13. As suggested above, if the authors decide to add the studies reporting unadjusted effect estimates. They may consider computing the measure of association for the studies where it is not reported but relevant data is available in the published paper or authors of the relevant studies could be contacted for data if possible.

Response: To set a minimum quality standard, we decided to restrict our analyses to studies that were at least adjusted for age.

Concern #33: 14. In the figures (forest plots and funnel plots), I can track only a maximum of 20 results pooled together, but in the results section 23 articles from 25 studies or even possibly 28, (while there are 34 records of studies in table 1). The number of studies and number results included in the pooled analysis should be consistent otherwise an explanation is needed in the text as to why there is a discrepancy.

Response: Please see response to concern #30. We have now added footnotes to clarify this to figure legends.

Concern #34: Table 1 and forest plots are confusing in terms of referring to a study name (with more than one references) rather results from the included published papers. A standard way of citation (Harvard reference, 1st author's surname et al., and year) would have been better. It does not matter if numerical numbers for references have been used in the main body. I am not quite sure why a study name is more appropriate than a reference (citation) to the published paper from which the measure of association has been extracted. Also, it would be better to arrange studies in the table by the alphabetical order of the author's names. These comments also apply to all forest plots in the figures. If for some of the studies, the authors had to use more than one articles, I think it would have been better to add the main article, which provide the measure of association in the tables and figures, and add a footnote to the tables that additional data is extracted from other articles with reference number X.

Response: We understand the reviewers point and can see the merits of presenting the studies by author names. However, there are several reasons we have chosen to present the studies according to cohort name. First, most of the cohorts included in the analysis are widely known by cohort name and thus, presenting results by cohort name is more informative for the reader. Secondly, the reviewer is correct in saying that we had to extract data from several articles for some cohorts and while we could prioritise one study over another for the purpose of the table, we believe this would not reflect contributions of the other papers and authors as well. Furthermore, the additional data from three sources of individual participant data has not already been published and therefore, does not have an author. Thus, we believe presenting the cohort names is the most appropriate choice both in terms of being informative for the reader and fair to the study authors whose manuscripts have contributed the data for this paper. However, for greater clarity around this we have a footnote to Table 1, as requested by the reviewer.

In regard to forest plots, we believe the same argument applies. Furthermore, for forest plots, we have ordered studies by effect size which makes plots more readable and informative. We are happy to take direction from the editor on this, if further discussion is required on this decision.

Concern #35: In table 1, a column on methods used in the study for assessing the association (e.g., main aim, study design, and statistical analysis, measure of association) would have been better. I suggest moving this table 1 to an eTable and giving its place in the main paper to a modified eTable2 as I suggest in the point below.

Concern #36: eTable 2, I suggest moving this table to the main paper and name it Table1 and add the effect estimates of each original study above the pooled estimates. Also, add explanation in the footnote to the table wherever it is required for various things, such as explaining what is meant by multiple adjustment, maximum adjustment etc.

Response to 35 and 36: We have now added eTable2 to the paper but we believe Table 1 is still an important part of the main paper so we have retained this in the main text.

We believe that an additional column of information with main study aim, study design, statistical analysis would be redundant as the main aim and study design is already stated given that all papers examined smoking and lung cancer and all included studies are cohorts. We also believe that a column on statistical analysis performed is unnecessary. While the suggestion of addition of the measures used in each study to Table 1 is a useful one, we are not able to do this as we did not extract this information.

The effect estimates of each original study are provided in the various forest plots.

We have now added footnotes to clarify the meaning of terms used in tables and figures. Multiple is anything that adjusted for more than just age. We considered the “maximum” to be the most adjustments provided each time. For some studies, this would have been age-adjusted whereas other studies adjusted for more factors than age only (i.e. multiple-adjusted). We have corrected the term “maximum adjustment” to “maximum available adjustment” in the text and figures.

Concern #37: eFigure 3: Instead of funnel plot a contour funnel plot will be more useful. It allows the statistical significance of study estimates to be considered, see (Peters et al., 2008) for more detail.

Response: This has now been done. Please see eFigure3 of the Supplementary Material.

Concern #38: Publication bias: no results from “Begg's test” OR “Egger's test” are included in the results section, at least a p-value could have been provided in the results section and also added under the funnel plot.

Response: A p value for the Begg's test has now been added to the results and underneath the funnel plot contained in the Supplementary Material (eFigure 3).

Concern #39: While I appreciate the author's efforts on quality assessment, I would like to see some acknowledgment of the limitations of NOQAS in the discussion section. A couple of useful references: Sanderson S, Tatt ID, Higgins JPT. Tools for assessing quality and susceptibility to bias in observational studies in epidemiology: a systematic review and annotated bibliography. *Int J*

Epidemiol 2007;36(3):666-676. doi:10.1093/ije/dym018 Katikireddi SV, Egan M, Petticrew M. How do systematic reviews incorporate risk of bias assessments into the synthesis of evidence? A methodological study. J Epidemiol Community Health 2015;69(2):189-195. doi:10.1136/jech-2014-204711 Viswanathan M, Berkman ND, Dryden DM, Hartling L. Assessing risk of bias and confounding in observational studies of interventions or exposures: further development of the RTI item bank (10). 2013. Available from: <http://www.effectivehealthcare.ahrq.gov/search-for-guides-reviews-and-reports/?pageaction=displayproduct&productid=1612> NOTE: Table 1 is especially helpful.

Response: We agree with the reviewer that quality assessment tools such as the Newcastle Ottawa Scale have their limitations. We have now mentioned this briefly in the discussion on page 15, lines 11-16 as follows: *“In addition, while we have aimed to assess study quality using the widely accepted and used Newcastle-Ottawa Scale, the value and contribution of quality assessment scales such as this to systematic reviews and meta-analyses continues to be debated. (10-12)”*

Concern #40: I suggest an extended table for the NOQAS having the following information: Selection: REC: Representativeness of exposed cohort (i.e., Is the exposed cohort ‘truly’ or ‘somewhat’ representative of exposed group) SNEC: Selection of the non-exposed cohort? (i.e., Is the non-exposed cohort drawn from the same community as the exposed cohort?). AE: Ascertainment of exposure (i.e., Is ascertainment based on either medical records or a structured interview?). DONPS: Demonstration that outcome of interest was not present at start of the study. Comparability: Design: Study controls for the most important factor (i.e., socio-economic and demographic status for cohort studies). Analyses: Study adjusts results for additional potential confounders Outcome: Assessment: Assessment of outcome (i.e., Is self-reported lung cancer validated by a medical test). LFU: Was follow-up long enough for outcome to occur? AFUC: Adequacy of follow-up of cohorts.

Response: We have now added the checklist used to the appendix, to clarify the assessment criteria used for quality assessment (see Appendix 3).

Concern #41. Further to the quality assessment, there is a lack of presenting the results. I think based on NOQAS it should be clearly explained, how the authors declared a study to be of a good quality? In addition, as table 1 describes the characteristics of the studies and is published in the main paper, it will be helpful to add the final NOQAS to this table.

Response: We have now added the results of the quality assessment to the text as follows on page 9 lines 28-30: *“Of 29 studies, four studies had a quality score of 5 out of 9, 9 studies had a score of 6, 12 studies had a score of 7 and 4 studies with a score of 8 (eTable 1).”* Furthermore, we have added the final results of the quality assessment to Table 1 as requested. In the appendix, we have also included the quality checklist and clearly indicated what criteria were used to score each study in the different domains. On page 7 of the appendix, we also state that *“Study quality assessment was based on the nine-star NOS using pre-defined criteria namely: selection (population representativeness), comparability (Adjustment for confounders), and ascertainment of outcome. The NOS assigns a maximum of four points for selection, two points for comparability and three points for outcome. Nine points on the NOS reflects the highest study quality.”*

Concern. I wondered, whether the women-to-men RRR of lung cancer would differ by the quality of the study, possibly NOQAS \geq 7 to be classed as of high quality.

Response: A subgroup analysis by study quality has now been added to the paper, as requested by the reviewer. This shows that the results from our analyses did not differ substantially by study quality.

Concern #43: On p.7, the ranges of prevalence of smoking and smoking cessation for women and men have been reported. It will be helpful if the prevalence can also be pooled separately for each sex and combined. This is important as the author argue that smoking has not reached its maturity in women. The pooled provenances of smoking could also be compared between the two sexes.

Response: We understand and see why the reviewer has highlighted this but analysis of prevalence was not the objective of the study and most studies did not report prevalence with confidence intervals or other data on variability necessary to allow meta-analysis to be performed. In any case, we have addressed the differing prevalence of smoking in women and men across studies in our sensitivity analyses in eTable 3.

Concern #44: Explaining the context of maturity of smoking in women, studies have shown that there was a greater disparity of smoking between women and men in oldest population, but this has decreased substantially in the youngest population (see peters et al 2014). In western population, “the women-to- men ever-smoking ratio ranged from 0.57 in the oldest to 0.87 in the youngest birth cohort”. If the trend continues, one may assume that there will be no difference. The results should be explained in the light of these results. Also, as they have got smoking provenance data for both sexes, they may be able to adjust the results for the interaction of smoking prevalence and sex, this may explain the difference if it exists.

Response: We agree with the reviewer that it is important to explain our findings in the context that eventually the women to men smoking prevalence may well be equivalent. We have performed subgroup analysis by the men to women smoking prevalence which showed no appreciable differences in study findings across these subgroups (Table 3). We have discussed this issue as follows on page 13, line 49 to page 14, line 16: *“Our results were consistent across regions and irrespective of the women-to-men smoking ratio, suggesting that underestimation of the association of smoking with lung cancer in women due to sex differences in smoking prevalence and underreporting of smoking is unlikely. This is especially relevant for parts of Asia where the prevalence of smoking in women is typically <10% and where smoking among women remains relatively socially unacceptable. As the up-take of smoking continues among women in countries where significant sex differences in smoking prevalence exist, the sex-specific risks of lung cancer due to smoking may become further apparent. This is also true in regards to Western countries where differences in prevalence between women and men have reduced substantially over time, with prevalence of smoking in younger cohorts of women and men approaching unity.(13) “*

Concern #45: There is a substantial amount of between-study heterogeneity, therefore meta-regression is inevitable, it is not explained why the author did not conduct meta-regression to explore

the sources heterogeneity using the study level covariates/moderators to understand these diverse findings. I strongly recommend the use of meta-regression; this might be helpful in explaining the sex differences. Also, as reported for some of the studies, subject level data is also available, if it is believed that there might be sex-related differences, I wondered if subject level data has been explored after the meta-analysis.

Response: We agree that meta-regression is a useful additional tool and have performed this to statistically test for heterogeneity across subgroups (Table 3).

The subject level data from the IPD did not allow for more detailed analyses than those presented. In particular, further studies are required to explore sex differences in the risk of histological subtypes of lung cancer, associated with smoking.

Concern #46: I think text on p.8 in the results section and related Table 2 need revision; it is difficult to understand what it means, also in the text below RRR and table 2 heading RR: "There was no evidence for a difference in the RRR according to the women-to-men ratio of current smokers ($P=0.90$) or the women-to-men ratio of lung cancer incidence in the studies ($P=0.64$) (Table 2)." The statistical analysis section does not explain clearly, what statistical methods was used for these analyses (results in Table 2). An explanation in the footnote under table 2 should be added.

Response: We have now clarified the text of this paragraph as follows on page 10, lines 13-26: "*The sex difference in the risk of smoking-related lung cancer in our main analysis did not differ in subgroup analyses stratified by the women-to-men ratio of current smokers ($P=0.90$), women-to-men ratio of lung cancer incidence in the studies ($P=0.64$), year of study baseline ($P = 0.66$), study endpoint ($P=0.21$) or study region ($P=0.73$) (Table 3). The sex difference in the risk of smoking-related lung cancer in our main analysis also did not differ by or follow-up time ($P=0.83$) or study quality ($P=0.69$). Furthermore, we have changed the heading of Table 2 (now Table 3) to improve the clarity of this. Random effects meta-analysis were used for all subgroup analyses and differences between subgroups were examined using meta-regression analyses. However, we have added this as a footnote to the table, as requested and also to the methods section to clarify any ambiguity.*

Concern #47: It is not clear in table 1 and text, how smoking has been assessed in the original studies. Some data on smoking (self-reported, or biochemically validated) could have been added to table 1. Similarly, assessment of the outcome of lung-cancer (self-reported or diagnosed/ascertained by a medical test) for each study need to be added to table 1.

Response: We, unfortunately, do not have these data available. However, the vast majority of studies would have had self-reported smoking data and medical diagnosis of lung cancer.

Concern #48: Discussion: 28. In the 1st paragraph of discussion the results are summarized well, and the authors' further explanation for no difference among women and men might be reasonable. This might need some published studies support (reference) to confirm that smoking is not matured among women. Currently, it seems like the explanation is based on speculation. On the basis of their

findings, they may recommend further research in a well-designed study, where it could be typically assessed whether any gender-specific difference in the outcome of lung cancer exist among smokers.

Response: A reference for this has now been added as requested by the reviewer. We also elaborate on the potential implications for future research in more detail as follows on page 15, lines 28-43: *“Further studies of the smoking-related risks of lung cancer in women and men are required as the smoking epidemic reaches its full maturity in women. In particular, given the later up-take of smoking in women, studies which allow sufficient lag time for lung cancer to develop are essential. In addition, reducing under-reporting of smoking in women, using standardised and robust methods for the ascertainment of smoking status and smoking behaviours and more extensive measurement and adjustment for confounders which differ by sex (such as exposure to environmental tobacco smoke) is also important.”*

Concern #49: As the study has extracted data from studies published globally, it might be useful if the difference of smoking related lung-cancer between women and men is explored among the high and low-income countries (if possible).

Response: We have performed two region-specific analyses, which we believe address the issue of high vs. low income country as best we can (see Table 3).

Concern #50: 30. p.10, 2nd paragraph “The relatively small relative risk estimates observed in this study are likely to partly reflect the heterogeneity in study populations in terms of baseline year of study, population age, prevalence of smoking, and smoking duration and intensity”: The author need to clarify, does “small relative risk estimates” refer to separate risk (RR) for women and men or women to men ratio of RR. This explanation of heterogeneity is not consistent with the results and statistical methods. The authors need to explore heterogeneity using Meta-regression.

Response: We have removed this statement as the reviewer is correct in saying that our explanation is not supported by the results of the subgroup analyses examining heterogeneity. We have performed meta-regression for several characteristics which are detailed in Table 3.

Concern #51: 31. Reference 1: “Institute for Health Metrics and Evaluation. Global Burden of Disease 2015. 2015. <http://vizhub.healthdata.org/gbd-compare/#>” need to be archived online.

Response: We are unsure what the reviewer means by this comment. However, we checked the link and it is still active.

1. Huxley RR, Peters SA, Mishra GD, Woodward M. Risk of all-cause mortality and vascular events in women versus men with type 1 diabetes: a systematic review and meta-analysis. *The Lancet Diabetes & Endocrinology*. 2015;3(3):198-206.

2. Yu Y, Liu H, Zheng S, Ding Z, Chen Z, Jin W, et al. Gender susceptibility for cigarette smoking-attributable lung cancer: A systematic review and meta-analysis. *Lung cancer*. 2014;85(3):351-60.
3. Lee PN, Forey BA, Coombs KJ. Systematic review with meta-analysis of the epidemiological evidence in the 1900s relating smoking to lung cancer. *BMC cancer*. 2012;12(1):385.
4. Devesa SS, Bray F, Vizcaino AP, Parkin DM. International lung cancer trends by histologic type: male: female differences diminishing and adenocarcinoma rates rising. *International journal of cancer*. 2005;117(2):294-9.
5. Nelson DE, Davis RM, Chrismon JH, Giovino GA. Pipe smoking in the United States, 1965–1991: prevalence and attributable mortality. *Preventive medicine*. 1996;25(2):91-9.
6. Melikian AA, Djordjevic MV, Hosey J, Zhang J, Chen S, Zang E, et al. Gender differences relative to smoking behavior and emissions of toxins from mainstream cigarette smoke. *Nicotine & Tobacco Research*. 2007;9(3):377-87.
7. Rothman KJ, Greenland S, Lash TL. *Modern Epidemiology*: Lippincott Williams & Wilkins; 2008.
8. Patrick DL, Cheadle A, Thompson DC, Diehr P, Koepsell T, Kinne S. The validity of self-reported smoking: a review and meta-analysis. *American journal of public health*. 1994;84(7):1086-93.
9. Jung-Choi K-H, Khang Y-H, Cho H-J. Hidden female smokers in Asia: a comparison of self-reported with cotinine-verified smoking prevalence rates in representative national data from an Asian population. *Tobacco Control*. 2012;21(6):536-42.
10. Sanderson S, Tatt ID, Higgins J. Tools for assessing quality and susceptibility to bias in observational studies in epidemiology: a systematic review and annotated bibliography. *International journal of epidemiology*. 2007;36(3):666-76.
11. Katikireddi SV, Egan M, Petticrew M. How do systematic reviews incorporate risk of bias assessments into the synthesis of evidence? A methodological study. *Journal of epidemiology and community health*. 2014;jech-2014-204711.
12. Viswanathan M, Berkman ND, Dryden DM, Hartling L. Assessing risk of bias and confounding in observational studies of interventions or exposures: further development of the RTI item bank. 2013.
13. Peters SA, Huxley RR, Woodward M. Do smoking habits differ between women and men in contemporary Western populations? Evidence from half a million people in the UK Biobank study. *BMJ open*. 2014;4(12):e005663.

VERSION 2 – REVIEW

REVIEWER	Joshua Muscat Penn State, USA
REVIEW RETURNED	22-May-2018
GENERAL COMMENTS	The authors have satisfactorily addressed the critiques.
REVIEWER	Tao Chen Liverpool School of Tropical Medicine, UK
REVIEW RETURNED	30-May-2018
GENERAL COMMENTS	This article is improved a lot with addressing the points from the reviewers. The limitations of this study have been clearly described.
REVIEWER	Dr Muhammad Riaz Department of Health Sciences, University of Leicester, UK.
REVIEW RETURNED	25-May-2018
GENERAL COMMENTS	Majority of my concerns are addressed satisfactorily and I believe the changes has improved the manuscript considerably. I recommend the paper to be accepted for publication. I do not have any further major comments, but following to author

	replies, I would like to comment as follow: Comment1: This comment is related to my previous comment#28 and #35: Statistical methods and the measures of association reported by a particular study would have been very helpful in terms of assessing and explaining the heterogeneity. Comment2: In response to my previous comment#29, I assume the author means “the sum of the variances of women and men specific log RRs” as described in the method section and not “the sum of women and men specific log RRs”. However, I have re-read the statement that relates to the standard error of log (RRRs) for each study and it is ok. Comment3: In clarification to my previous comment#51: In a manuscript submitted for publication, if a reference is a link to a webpage, it is required to be archived. This allows that the information contained in the webpage will be accessible in the archive even if it is removed from the website. This comment can be ignored if it is not a requirement for BMJ. Information of archiving a website can be found via link https://nationalarchives.gov.uk/documents/information-management/web-archiving-guidance.pdf comment4 p.21, table 1, last column: "available" is used twice, delete as appropriate. comment5 p.25, table 3, "P for interaction", explain in the footnote, how the interaction is assessed.
--	---

VERSION 2 – AUTHOR RESPONSE

Reviewer: 1

Reviewer Name: Joshua Muscat

Institution and Country: Penn State, USA

Please state any competing interests or state ‘None declared’: None

Comment: The authors have satisfactorily addressed the critiques.

Reviewer: 5

Reviewer Name: Dr Muhammad Riaz

Institution and Country: Department of Health Sciences, University of Leicester, UK.

Please state any competing interests or state ‘None declared’: None

Comment: Majority of my concerns are addressed satisfactorily, and I believe the changes has improved the manuscript considerably. I recommend the paper to be accepted for publication. I do not have any further major comments, but following to author replies, I would like to comment as follow:

Concern #1: This comment is related to my previous comment#28 and #35: Statistical methods and

the measures of association reported by a particular study would have been very helpful in terms of assessing and explaining the heterogeneity.

Response: Unfortunately, while we understand the reason for the reviewer's request and agree that these would be useful additions, we have not recorded the statistical methods used in each study or type of measure of association in each study. However, most measures of association were hazard ratios or risk ratios as all included studies were cohort studies and thus, the statistical methods used are not vastly different across studies due to this.

Concern #2: In response to my previous comment#29, I assume the author means "the sum of the variances of women and men specific log RRs" as described in the method section and not "the sum of women and men specific log RRs". However, I have re-read the statement that relates to the standard error of log (RRRs) for each study and it is ok.

Response: Thank you.

Concern #3: In clarification to my previous comment#51: In a manuscript submitted for publication, if a reference is a link to a webpage, it is required to be archived. This allows that the information contained in the webpage will be accessible in the archive even if it is removed from the website. This comment can be ignored if it is not a requirement for BMJ. Information of archiving a website can be found via link <https://nationalarchives.gov.uk/documents/information-management/web-archiving-guidance.pdf>

Response: This is not a requirement for publication in BMJ. Therefore, as advised by the reviewer we have not archived the reference.

Concern #4: p.21, table 1, last column: "available" is used twice, delete as appropriate.

Response: This has now been amended.

Concern #5: p.25, table 3, "P for interaction", explain in the footnote, how the interaction is assessed.

Response: The P for interaction was assessed using meta-regression and this has now been added as a footnote to the table, as requested.

Reviewer: 4

Reviewer Name: Tao Chen

Institution and Country: Liverpool School of Tropical Medicine, UK

Please state any competing interests or state 'None declared': None declared

Comment: This article is improved a lot with addressing the points from the reviewers. The limitations of this study have been clearly described.